# Wireless magneto-ionics: voltage control of magnetism by bipolar electrochemistry

Zheng Ma [1,6], Laura Fuentes-Rodriguez [2,3,6], Zhengwei Tan [1], Eva Pellicer [1], Llibertat Abad [3], Javier Herrero-Martín [4], Enric Menéndez [1]✉, Nieves Casañ-Pastor [2]✉ & Jordi Sort [1,5]✉

Modulation of magnetic properties through voltage-driven ion motion and redox processes, i.e., magneto-ionics, is a unique approach to control magnetism with electric field for low-power memory and spintronic applications. So far, magneto-ionics has been achieved through direct electrical connections to the actuated material. Here we evidence that an alternative way to reach such control exists in a wireless manner. Induced polarization in the conducting material immersed in the electrolyte, without direct wire contact, promotes wireless bipolar electrochemistry, an alternative pathway to achieve voltage-driven control of magnetism based on the same electrochemical processes involved in direct-contact magneto-ionics. A significant tunability of magnetization is accomplished for cobalt nitride thin films, including transitions between paramagnetic and ferromagnetic states. Such effects can be either volatile or non-volatile depending on the electrochemical cell configuration. These results represent a fundamental breakthrough that may inspire future device designs for applications in bioelectronics, catalysis, neuromorphic computing, or wireless communications.

Precise control over selected electronic, magnetic, chemical and/or structural properties of materials is required for a wide range of applications such as batteries[1], fuel cells[2] and information storage or computing[3,4]. Recently, the use of room-temperature ionic transport to modify magnetic properties (i.e., magneto-ionics) has gained much interest in the pursuit of voltage control of magnetism for energy-efficient spintronics. Under application of external voltage, the electrochemistry fundamental aspects behind the formation of electric double layer (EDL) at the interface between the solid target material and an adjacent electrolyte, as well as the redox changes occurring at the ionic/electronic conducting material, are central to enable ionic migration. Due to their very narrow thickness (<0.5 nm), EDLs lead to very strong electric fields under gate voltages of a few Volt[5]. In semiconductors, this causes accumulation of ions at their channelled surface[6,7]. When electrochemical reactions occur, discharge takes place through charge transfer, due to the mixed ionic-electronic conductivity, eventually modifying the entire material structure. Magneto-ionics has been explored using electrolyte gating in either transistor- or capacitor-like device configurations[8] as a means to toggle several parameters, such as perpendicular magnetic anisotropy[9], magnetization[10,11], exchange bias[12] and domain wall motion[13]. As an extreme case, magneto-ionics can trigger room-temperature reversible paramagnetic-to-ferromagnetic transitions (ON-OFF ferromagnetism), in, for example, electrolyte-gated oxide and nitride films of cobalt[14]. Remarkably, nitrogen ion transport tends to occur uniformly creating a plane-wave-like migration front which is highly beneficial to boost cyclability[14].

[1]Departament de Física, Universitat Autònoma de Barcelona, 08193 Cerdanyola del Vallès, Spain. [2]Institut de Ciència de Materials de Barcelona, CSIC, Campus UAB, 08193 Bellaterra, Barcelona, Spain. [3]Institut de Microelectrònica de Barcelona-Centre Nacional de Microelectrònica, CSIC, Campus UAB, 08193 Bellaterra, Barcelona, Spain. [4]ALBA Synchrotron Light Source, 08290 Cerdanyola del Vallès, Spain. [5]Institució Catalana de Recerca i Estudis Avançats (ICREA), Pg. Lluís Companys 23, Barcelona 08010, Spain. [6]These authors contributed equally: Zheng Ma, Laura Fuentes-Rodriguez. ✉e-mail: enric.menendez@uab.cat; nieves@icmab.es; jordi.sort@uab.cat

In magneto-ionic devices, electrodes are usually grown adjacent and in direct contact to the target material. Bias voltages to induce ion diffusion are generated by directly connecting these electrodes to a power supply using electrically conductive wires. This standard way to apply voltage is suitable for applications where a physical connection to the actuated material is not a drawback. However, in many cases, such as biomedical stimulation, microfluidics, magnonics or remotely actuated magnetic micro/nano-electromechanical systems, it might be desirable to induce magneto-ionic effects in a wireless manner.

Interestingly, it has been shown that the surface of electrically conducting objects immersed in liquid electrolytes can become polarized under the action of external electric fields yielding to electrochemical processes. This phenomenon is called "bipolar electrochemistry" (BPE) since it leads to the formation of a dipole with induced anode and cathode poles in the immersed object, along the electric field direction and opposing the external field, where electrochemical reactions may occur for certain potentials without any direct wire connection. In recent years, BPE has gained renewed attention for electrosynthesis of novel materials[15,16], and for its potential in applications such as sensing, screening and biological actuation[15,17,18], involving processes not only at the surface but also within the material if intercalation/deintercalation and ionic motion does exist. The wireless voltage-control of physical properties, and specifically magnetism, through BPE has not been explored yet, although it would be of highly relevant interest that offer unexplored perspectives and new engineering options.

In this work, we demonstrate the ON-OFF switching of ferromagnetism in cobalt nitride (CoN) via wireless magneto-ionics. Compared to standard electrolyte-gating methods, here the magneto-ionic material is not directly wired to external power sources. Instead, an electric dipole is induced wirelessly on the target material under the action of an electric field created by external electrodes immersed in the electrolyte medium, leading to chemical processes at the induced poles (i.e., BPE). Modulation of magnetism can be made temporary (volatile) or permanent (non-volatile) depending on the device configuration with respect to the external field. Redox gradients induced in the immersed magneto-ionic material in the horizontal configuration lead to dynamical redox/ionic processes that result in temporary ferromagnetism. Conversely, the vertical configuration (where the magneto-ionic sample is placed parallel to the driving electrodes) develops chemical changes that turn into a permanent ferromagnetic signal. Beyond magneto-ionics, the reported approach could be used to tune other voltage-dependent physical/chemical properties of materials such as superconductivity[19,20] or metal-insulator transitions[21], and it is likely to open new avenues in iontronics and wireless magnetoelectric devices.

## Results

### Wireless magneto-ionics in horizontal configuration

Figure 1a illustrates a home-made bipolar electrochemical cell setup with horizontal configuration, where a pair of parallel vertical Pt plates are used as external driving electrodes that generate the electric field, and a 50-nm CoN thin film grown on top of Au (60 nm)/Ti (20 nm)/Si acts as the immersed conducting material where polarization effects are to be induced without any electrical wiring. In this experimental scheme, the films are immersed in the electrolyte solution (0.1 M KI in propylene carbonate, PC) and aligned horizontally between the Pt electrodes. Photographs of the experimental setup for the horizontal and vertical configurations are shown in Sections 1 and 2 of the Supplementary Information. Voltages applied to the Pt electrodes generate potentials at opposite poles of the CoN layer, which may yield to capacitive effects (EDL, see Fig. 1a) and, if sufficiently high, to chemical reactions at the induced anode and cathode poles of the CoN sample. Voltammetry scans performed ex-situ and through direct contact to the sample indeed show that the CoN coating undergoes chemical

reduction, pointing to the formation of species with several Co:N stoichiometries (Section 3, Supplementary Information). Such redox processes are likely to occur here also through wireless induction of poles. Note that neither visible chemical reactions nor magnetic properties changes are observed upon immersion of the CoN in the electrolyte with the absence of external applied voltages (Section 4, Supplementary Information).

Figure 1b depicts the potential profiles for electrolyte voltage drop, the distortion created for the immersed conducting material, and the resulting induced potentials at the poles. The interfacial potential difference between the CoN and the electrolyte solution, which is the driving force of electrochemical reactions, varies in this configuration along the lateral length of the actuated film[22,23]. As revealed by COMSOL simulations (Section 5, Supplementary Information), the induced potentials are the highest at the edges of the sample, where anodic and cathodic poles form. Consequently, electrochemical processes are always observed there first. Unfortunately, a direct measurement of the induced dipole is not possible through direct contact, since the dipole discharges, but it was evaluated previously using an indirect approximation[24], and the results agree with the simulation shown here through finite element methods (see Fig. S5). This indirect approximation consisted of connecting one pole to one edge of the sample while leaving the other measuring pole in the nearby electrolyte, which prevents discharge. Although the ionic mobility and diffusion generate noise, the distortion of the potential profile across the sample could be still observed in aqueous media[24]. The values previously obtained were in qualitative agreement with electrostatic simulations, thereby confirming the validity of such simulations. The same type of experimental evaluation is not feasible using the PC electrolyte within the potential window available and for the dimensions of the samples prepared here, and therefore the COMSOL calculations are shown instead (see Fig. S5).

Room-temperature hysteresis loops measured ex situ by vibrating sample magnetometry (VSM) for the sample before and after applying an external driving voltage of 15 V for 5 min in the horizontal configuration are presented in Fig. 1c. While a paramagnetic state ("OFF" ferromagnetism) is evidenced from the virtually zero net magnetization for the pristine film, a clear ferromagnetic hysteresis loop ("ON" ferromagnetism) with a maximum magnetization, $M_S$, of 53 emu cm$^{-3}$ builds up for the wireless voltage-actuated samples. Notably, consecutive hysteresis loops measurements show that $M_S$ significantly drops in magnitude over time at ambient conditions. $M_S$ reduces by more than 50% within 12 hours and decreases below 10 emu cm$^{-3}$ after around 24 hours (a value which is <0.7% the saturation magnetization of metallic Co), evidencing that this process is virtually volatile. This magnetization depletion is most likely related to the existence of a charge gradient within the material that facilitates redistribution of charges and ions concentrations from the reduced cathode towards the rest of the sample. Such kind of gradients have been observed before in several BPE systems, for the same configuration[16,25,26]. The hysteresis loop measurements on the individual negative and positive poles and the central part, split from a sample after the bipolar treatment, confirm a significant gradient in the corresponding $M_S$ with larger $M_S$ towards the positive pole (Section 6, Supplementary Information). On the basis of these results, we hypothesize that, upon the removal of the external positive Pt pole, the charge gradient created along the sample and the corresponding redox and ionic changes promote internal diffusion of ions that equalize electronic oxidation states, resulting in an internal electrochemical discharge relaxation that restores electroneutrality[25].

Despite the fact that the generation of a net ferromagnetic response through the wireless method was proven, subsequent structural characterization of the treated samples was hindered due to the time evolution of the gradient observed in $M_S$ and therefore the derived chemistry, and the impossibility to perform all experiments in

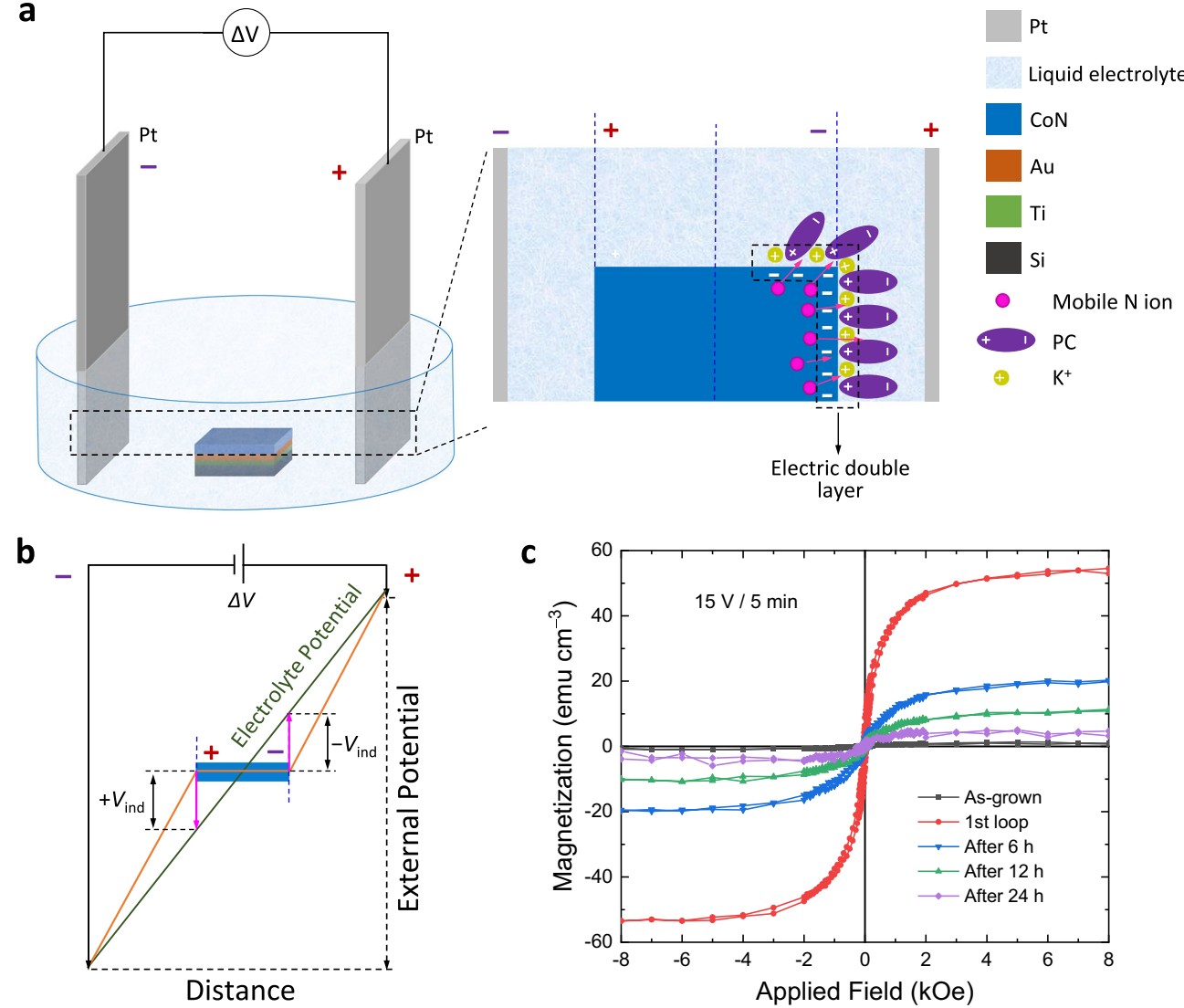

**Fig. 1 | Demonstration of wireless magneto-ionics in CoN films using an experimental setup with horizontal configuration. a** Schematic illustration of the electrochemical cell and a simplified sketch depicting the induced poles formed and the voltage-controlled nitrogen ion motion at the film-electrolyte interface. The two edges of the bipolar magneto-ionic sample, indicated by blue dashed lines, correspond to the anodic (left) and cathodic (right) poles, whereas the blue dashed line depicted at the centre of the sample corresponds to an initial zero-charge position where the interfacial potential difference is zero with respect to the electrolyte. Note that, for clarity, the sketch depicts only the reactions at the cathodic pole (a correlated I⁻ oxidation to I₂ occurs simultaneously at the solution near the induced anode). **b** Schematics of the potential profile for the electrolyte and the distortion occurring when a conducting material is immersed in the electrolyte. The deviation from the original electrolyte profile corresponds to the induced poles, opposing the external field, at the extremes of the material. **c** Room-temperature hysteresis loops for the CoN films subject to an external driving voltage of 15 V for 5 min using the setup shown in **a**. The evolution of the saturation magnetization over time indicates the volatility of the magneto-ionic effect for such configuration.

the same time period and therefore, that is, due to the volatile nature of the induced effects in the horizontal BPE treatments. It was then envisaged that the use of a vertical configuration, where all regions of the surface are equidistant from the external Pt parallel electrodes (and therefore subject to the same induced pole charging) might be a good strategy to achieve non-volatile (i.e., permanent) magneto-ionic effects.

## Wireless magneto-ionics in vertical configuration

Figure 2a shows that the magneto-ionic effect can also be achieved using a vertical BPE configuration. Here, the films are immersed in the liquid electrolyte (0.1 M KI in PC) and placed parallel to the Pt electrodes. An induced negative potential is generated on the CoN film, which is facing the driving Pt electrode charged positively (while a positive pole is created on the Pt sheet attached to the back of the

sample and facing the negative driving Pt electrode). Such negative potential promotes the formation of the EDL at the negative pole and nitrogen ionic motion towards the interface with the electrolyte, along with redox changes. In contrast to the horizontal BPE cell, here the interfacial potential difference is uniform along the film surface. As before, the ionic movement is correlated with an electrochemical redox reaction, namely a change in Co oxidation state, as it will be described below. In turn, the electrolyte surrounding the positive pole also undergoes a chemical reaction, where I⁻ is oxidized at the induced anode, forming I₂ or I₃⁻, leading to the observed orange colour in the electrolyte (Section 2, Supplementary Information). Simultaneous secondary reactions are possible in both poles at the largest applied potentials. For example, under the application of 15 V for more than 5 min, it is observed that bubbles start to appear on the induced cathode of the bipolar electrode surface, which could be related to the

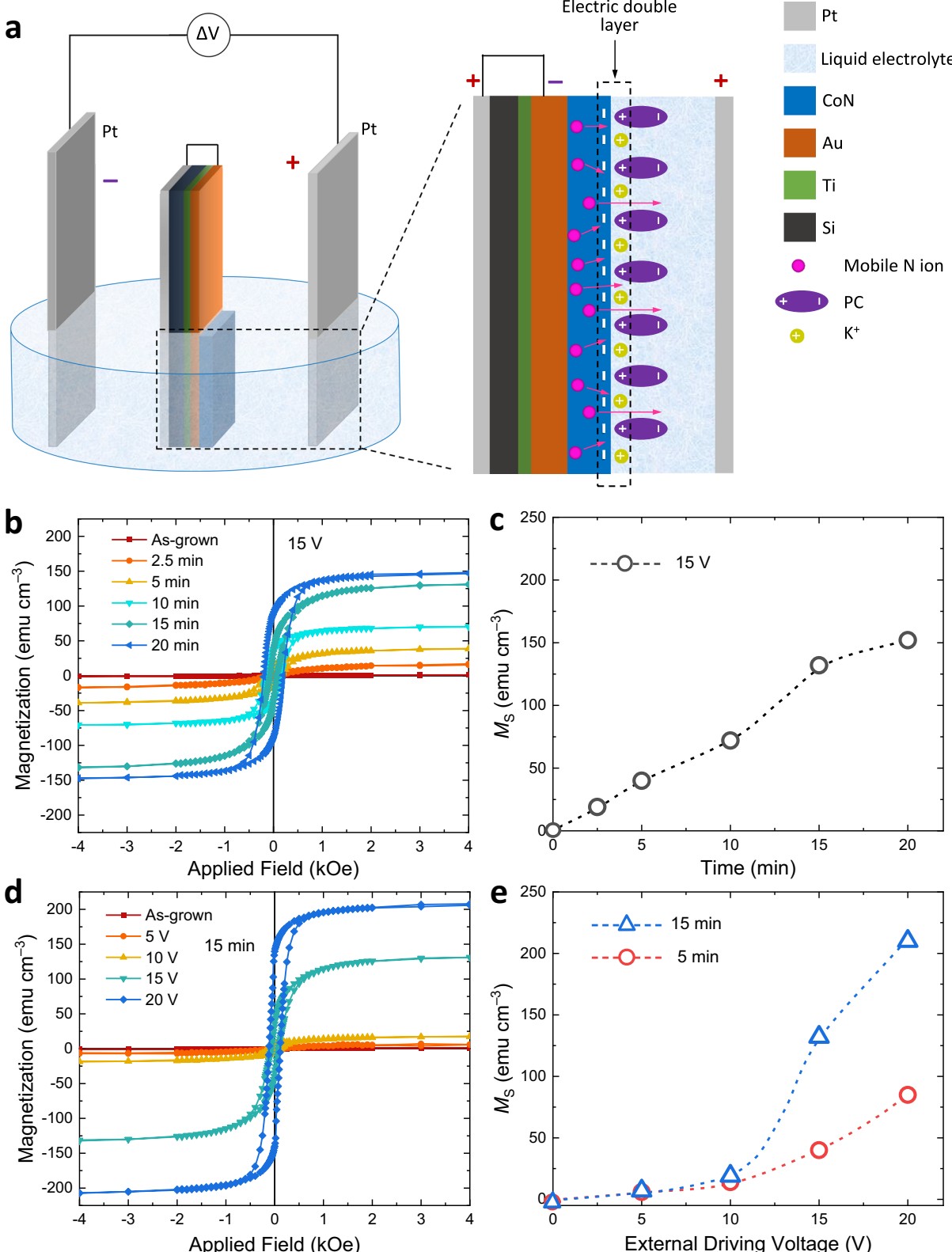

**Fig. 2 | Wireless magneto-ionic control of ON-OFF ferromagnetism at room temperature using a vertical bipolar electrochemistry cell. a** Schematic illustration of the electrochemical cell and a simplified sketch depicting the generation of the EDL and voltage-controlled nitrogen ionic motion at the film-electrolyte interface. Note that an electronically conducting path is created externally (shown as a bracket in the figure) between the Au underlayer supporting the CoN and a Pt sheet placed at the backside that acts as induced anode, to assure the creation of the induced dipole. Since Au is deposited on a Si wafer, no electric contact holds without such connection. **b** Room-temperature magnetic hysteresis loops for the as-grown and voltage-actuated films under a 15 V biasing voltage for various durations. **c** Saturation magnetization ($M_S$) vs. biasing duration upon applying 15 V. **d** Room-temperature hysteresis loops as a function of the magnitude of the driving voltage for the same biasing duration of 15 min for the pristine (as-sputtered) and voltage-actuated films. **e** $M_S$ versus external driving voltage at two different application times of 5 min (open red circles) and 15 min (open blue triangles). The dashed lines in **d** and **e** are guides to the eye.

formation of $N_2$ gas after reduction of CoN, or to the complex reduction of the propylene carbonate electrolyte to aliphatic species[27]. Since the connected Pt cathode also shows gas evolution the most probable reaction is derived from the solvent. Alternative secondary reactions, like Co metal oxidized by $I_2$ yielding soluble green $CoI_2$, do not seem to take place, since no Co depletion occurs (i.e., no bluish colour is observed and $M_S$ increases). A control experiment was carried out to verify this using metallic cobalt powder in a $I_2$-containing PC electrolyte (see Section 4, Supplementary Information). No iodine signal is observed either in the solid material, so no additional phases containing iodine form, in agreement with previous works using this electrolyte[28]. Open circuit potentials and initial voltammetry potentials are very stable and constant, evidencing no spontaneous direct reactions in the setting.

Voltage-driven changes in the magnetic properties are revealed by VSM measurements. Figure 2b shows room-temperature hysteresis loops for the as-grown sample and for samples after the vertical BPE treatment under a constant applied voltage of 15 V during various time spans. While no ferromagnetic signal is detected for the pristine film, a clear hysteresis loop with maximum magnetization, $M_S$, of 20 emu cm$^{-3}$ appears upon 15 V biasing for 2.5 min. By extending the voltage application time, the hysteresis loops become more square-shaped, and $M_S$ progressively increases, reaching 145 emu cm$^{-3}$ after 20 min (Fig. 2c). The dependence of the induced $M_S$ as a function of the driving voltage is shown in Fig. 2d for a fixed actuation time of 15 min. For 5 V and 10 V, hysteresis loops just start to emerge gradually, and $M_S$ remains lower than 25 emu cm$^{-3}$. A clear increase of $M_S$ is observed at 15 V, where $M_S$ reaches 130 emu cm$^{-3}$, suggesting that at this potential suitable conditions for the reaction are reached. $M_S$ exceeds 200 emu cm$^{-3}$ for a driving voltage of 20 V, the maximum external potential applied in the present study. With time and voltage, the variation of coercivity ($H_C$) shows no clear trends (values ranging from around 50 to 200 Oe), whereas the evolution of squareness ratio (ratio between remanent saturation $M_R$ and saturation magnetization $M_S$: $M_R/M_S$ (%)) tends to increase with both time and voltage (going from 7 to 66 %), as shown in Table S1 in the Supplementary Information. This is consistent with an increased amount of ferromagnetic counterpart (i.e., the evolution with time and voltage of $M_S$), with a better defined in-plane magnetic anisotropy. While the ferromagnetic behaviour becomes more pronounced for larger potentials, secondary reactions eventually detach the CoN coating above 20 V. The hysteresis loops as a function of driving voltage for an application time of 5 min are shown in Section 2, Supplementary Information. The evolution of $M_S$ as a function of biasing voltage for actuation times of 5 and 15 min is summarized in Fig. 2e. Between 10 and 15 V, a pronounced slope change occurs, evidencing that 15 V is above the onset voltage for magneto-ionics in this system. The maximum $M_S$, 210 emu cm$^{-3}$, obtained at the 20 V/15 min conditions is smaller than that in previous studies on CoN films actuated through direct electrical connections ($M_S \approx 637$ emu cm$^{-3}$)[14]. This difference is partially because, during the BPE experiment, the voltage induced at the film is much smaller than the externally applied driving voltage (e.g., the film is under 1 V when the Pt electrodes are biased at 15 V in the vertical BPE devices, as discussed in Section 5 of the Supplementary Information). Other contributing factors are the larger potentials used in the reported direct-wiring case, where the gating voltage was 50 V, and the use of different electrolyte media[28].

In contrast to the horizontal BPE, the resulting ferromagnetism using the vertical configuration is stable over time, as shown in Section 2 of the Supplementary Information. This allows for subsequent examination of the structural properties in these films. The volatility of the effect in the horizontal configuration was the occurrence of a redox/ionic lateral gradient within the zone where potentials are sufficiently high. Such gradient is likely to be present also for the vertical configuration (perpendicular to the sample), but with much smaller magnitude since the film is very thin in the field direction and now the entire outer surface of the film (and not only the edges) contributes to the redox process. Finally, the reversibility characterization of the magnetic change (Section 7, Supplementary Information) suggests the existence of quasi-reversibility for short treatment times (5 min), evidencing that a large fraction of the sample recovers within such time scale, possibly through redistribution of the remaining N ions.

Microstructural and compositional characterizations were conducted by high-angle annular dark-field scanning transmission electron microscopy (HAADF-STEM) and electron energy loss spectroscopy (EELS) on cross-section lamellae of the as-grown and voltage-actuated (treated at 15 V for 15 min in the bipolar vertical configuration) samples (Fig. 3). Both the untreated and treated CoN films show a fully dense structure and a flat surface. In contrast to the pristine film, where homogeneous distributions of Co and N elements are found, the treated film shows a two-layered morphological feature, with a nitrogen-depleted top layer that can be resolved from the elemental mapping. This evidences a redistribution of nitrogen ions and a uniform nitrogen ion migration front along the film perpendicular direction. The propagation of this front is accompanied by nitrogen ions release to the electrolyte. This effect, which is responsible for the gradual increase of ferromagnetic signal, is driven by the potential gradient occurring during the bipolar electrochemical process. Such planar ion migration front was also observed in electrolyte-gated magneto-ionic nitrides using conventional wired electrodes[14,29,30].

Based on previous literature on cobalt nitride with variable nitrogen concentration[31], the X-ray photoemission spectroscopy (XPS) study (Section 8, Supplementary Information) shows a relative change on the N/Co ratio and on the N components for the treated sample, thus corroborating the voltage-induced nitrogen ion release to the electrolyte. The EELS spectrum acquired from the bottom sublayer of the treated film closely resembles that of the pristine sample (Fig. 3c). For the top sublayer, however, the spectrum changes considerably with a significant increment in the relative intensity of Co $L_3$ white line compared to that of the as-prepared sample. The intensity ratio, $L_3/L_2$, is enhanced from around 2.6 for the latter to nearly 3.1 for the top sublayer (Fig. 3d). This indicates a decrease of Co valence states in the most affected top layer of the treated sample[32], resulting from the reduction process in the CoN film.

Aside from metallic cobalt, other Co–N interstitial compounds such as $Co_3N$ and $Co_4N$ ($Co^{2+}/Co^{3+}$ mixed valences with lower N/Co ratios) reportedly exhibit ferromagnetic behaviour at room temperature[33,34]. Therefore, the electrochemical redox reactions on the bipolar electrode involving ion deintercalation may give rise to reduced $CoN_{1-x}$ or Co magnetic species, which account for the observed voltage-induced magnetization changes. Figure 3e, f presents the high-resolution TEM images of the regions close to the film surface. The corresponding fast Fourier transform (FFT) is shown in Fig. 3g, h. The cubic CoN (200) and (111) reflections appear rather broad in the FFT pattern, which reveals the low-crystallinity nature of the as-grown film. For the voltage-actuated film, besides signals from CoN, the spots corresponding to an interplanar spacing of 2.38 Å (red circles in Fig. 3h) agree well with the (110) reflection of orthorhombic $Co_2N$ (*Pnnm*) phase. Furthermore, the blue circles in Fig. 3h (interplanar spacing of 1.91 Å) reveal the existence of hexagonal Co and/or $Co_3N$, in agreement with EELS. The coexistence of several phases with different stoichiometries intergrown during reduction was already suggested by linear seep voltammetry measurements (Section 3, Supplementary Information) that evidenced a second reduction wave after the initial reduction. These measurements, carried with direct electrical contact to the CoN sample, suggest that analogous electrochemical processes occur during wired and unwired magneto-ionic actuation.

To gain more insight into the voltage-driven phase transitions, X-ray absorption spectroscopy (XAS) and X-ray magnetic circular

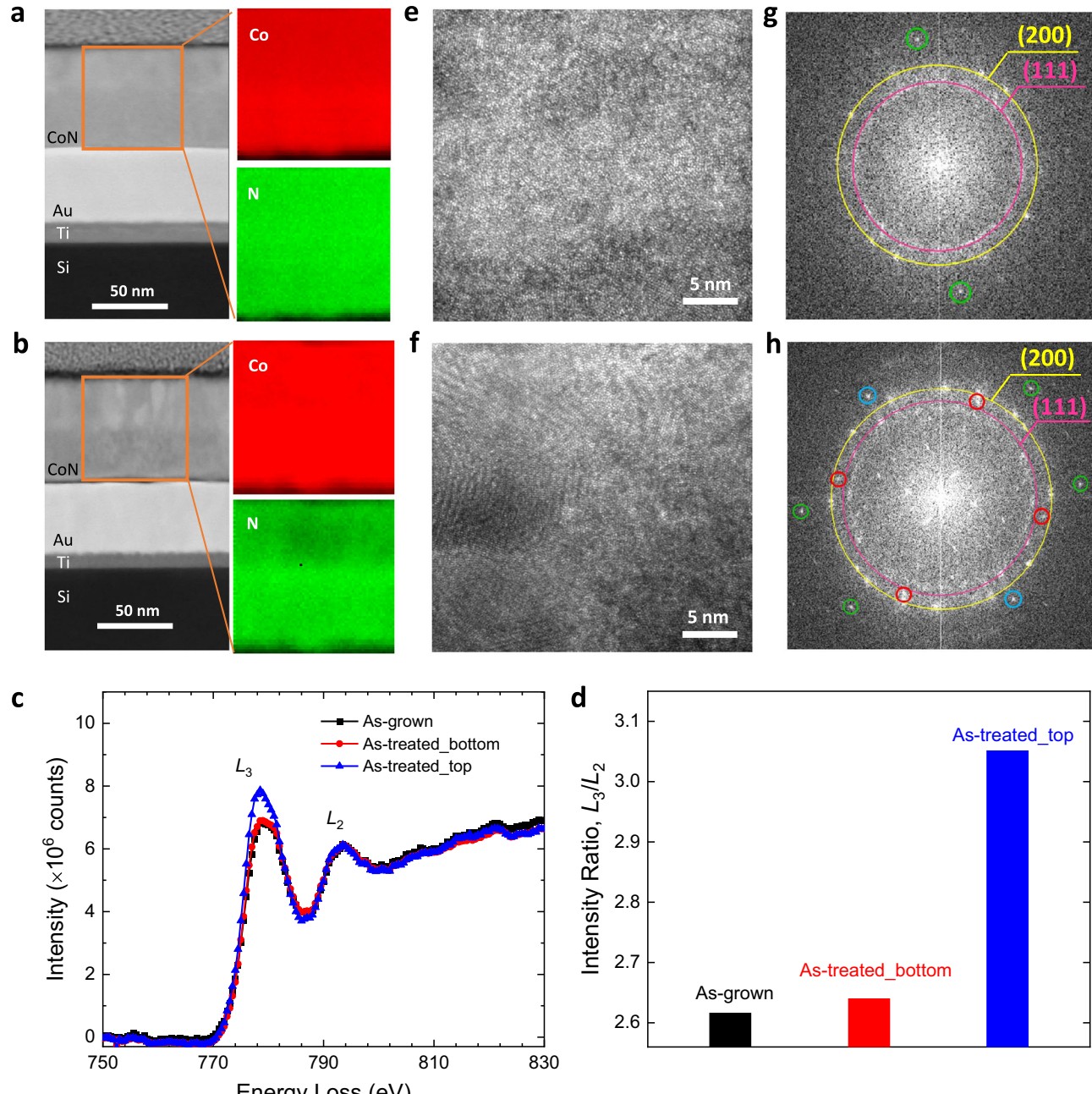

**Fig. 3 | Transmission electron microscopy (TEM) and electron energy loss spectroscopy (EELS) characterisation of the as-grown and as-treated CoN films. a**, **b** High-angle annular dark-field (HADDF) scanning TEM images and the corresponding elemental EELS mappings of the areas enclosed in orange of the as-prepared film, **a**, and the one subjected to 15 V for 15 min. **b** Red and green colours correspond to Co and N elements, respectively, for the EELS analysis. **c** EELS spectra acquired from the as-prepared film (black) and the two different regions of the treated film, the bottom layer (red) and the top layer of lean nitrogen (blue). The spectra are normalized to the $L_2$ line. **d** The calculated intensity ratios of the white lines, $L_3/L_2$. **e**, **f** High-resolution TEM (HRTEM) images of the as-grown film, **e**, and the film actuated at 15 V for 15 min. **f** The corresponding fast Fourier transform patterns are shown in **g** and **h**, respectively. The broad rings in yellow and pink correspond to the (200) and (111) reflections of cubic CoN. The spots in green match the CoN (220) reflection. The spots corresponding to a lattice spacing of 2.38 Å (red circles) agree well with the (110) reflection of orthorhombic $Co_2N$ (*Pnnm*) phase. The blue circles, corresponding to an interplanar spacing of 1.91 Å, are consistent with the hexagonal Co (101) or hexagonal $Co_3N$ (201) orientation.

dichroism (XMCD) measurements were performed. Figure 4a shows the Co $L_{2,3}$-edge XAS spectra of the as-grown film and the films treated under driving voltages of 15 V and 20 V for 15 min in the vertical configuration. The spectrum for the pristine sample strongly resembles that of low-spin $Co^{3+}$ XAS line shapes, which possess a prominent peak around 779 eV with a high-energy shoulder structure at the $L_3$ edge. This ascertains the primary $Co^{3+}$ valence state. Interestingly, the spectra of the actuated films exhibit three noticeable features (A, B,

and C on the right panels of Fig. 4). Specifically, in contrast to the spectrum of the pristine state, an additional peak is detected at around 776 eV for the treated samples (Fig. 4, feature A). This is the characteristic pre-peak of $Co^{2+}$. Additionally, there is an energy shift of the $L_3$-edge maximum towards lower energies upon voltage application (Fig. 4, feature B). For example, the $L_3$-edge maximum shifts from 779 eV for the pristine state to 778 eV for the sample actuated under 20 V. Similar behaviour is also visible for the Co $L_2$-edge (feature C in

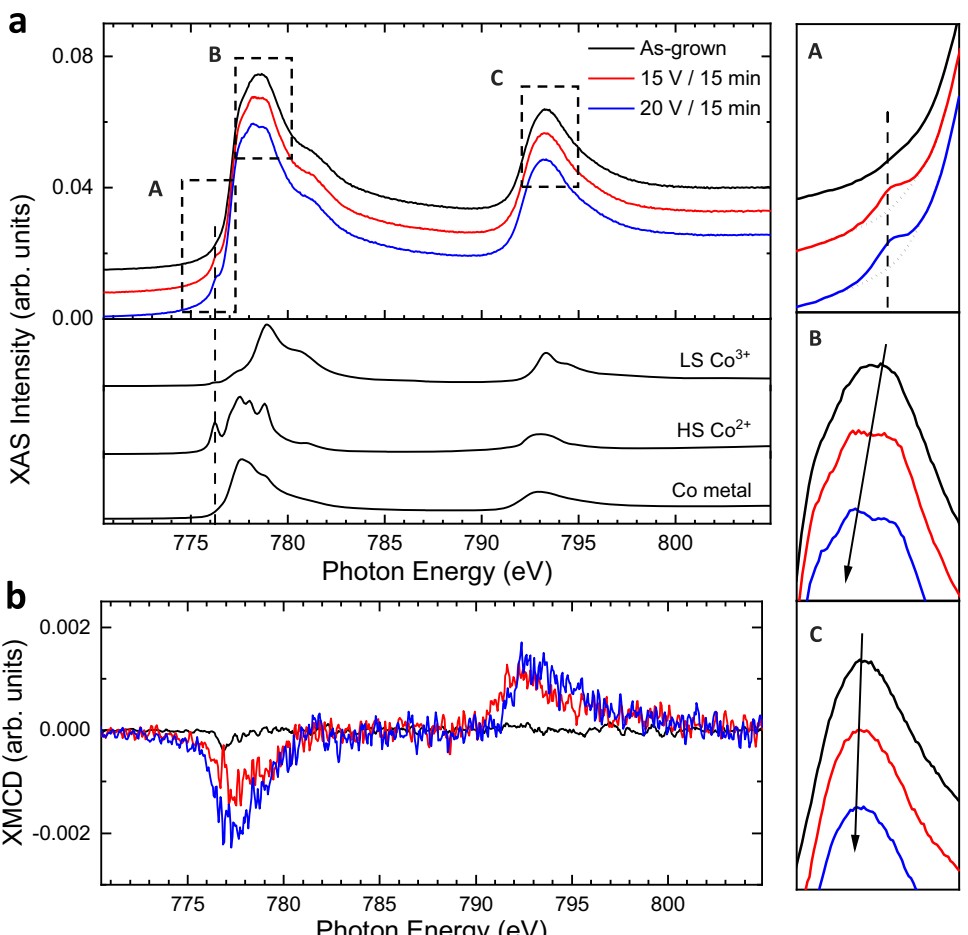

**Fig. 4 | Wireless magneto-ionic control of magnetism probed by soft X-ray absorption spectroscopy (XAS) and X-ray magnetic circular dichroism (XMCD).** Co $L_{2,3}$-edge XAS spectra, **a**, and XMCD, **b**, of the as-sputtered and actuated (applying driving voltages of 15 V and 20 V for 15 min) CoN films, represented by black, red and blue lines, respectively. The reference spectra of low-spin (LS) Co$^{3+}$, high-spin (HS) Co$^{2+}$, and metallic Co are shown in the bottom panel of **a**. The three panels on the right show enlarged views of the dashed rectangular areas, A, B, and C, in **a**. All XAS spectra are vertically shifted for clarity. The vertical dashed lines mark the appearance of reduced Co$^{2+}$ peak. The black downward arrows denote the shift of Co $L_3$ maximum.

Fig. 4). The shifting is related to the partial reduction of Co$^{3+}$ to lower oxidation states. Further, we find that the XMCD signal is practically zero for the as-grown sample (Fig. 4b), as expected. The treated samples, however, possess gradually increased XMCD intensities, resulting from a net magnetic moment from Co or CoN$_{1-x}$ species. This agrees with the previous magnetometry results. Taken together, our XAS-XMCD studies corroborate the Co valence transition from Co$^{3+}$ to Co$^{2+}$ and occurrence of metallic Co, which are responsible for the wirelessly induced magnetic response.

## Outlook

This work constitutes the experimental demonstration of wireless magneto-ionic effects in thin films via bipolar electrochemistry. The study is performed using CoN films immersed in a liquid electrolyte. Evidence for nitrogen ion motion and the associated electronic structure transitions has been obtained through microscopic and spectroscopic methods. In the absence of direct wired connections to the actuated material, voltage-induced transitions from paramagnetic to ferromagnetic states are driven by the formation of induced charged poles in the material in response to the externally applied voltage and the concomitant discharging processes (i.e., capacitive and faradaic components). We demonstrate that the ion-motion driven magnetic switching behaviour through BPE can be volatile or non-volatile depending on the chosen experimental configuration with regard to the imposed external electric field: for the vertical setup, the

uniform charging and corresponding redox processes on the CoN film give rise to sizable magnetization changes with high stability, whereas volatile magneto-ionic effects are observed for the horizontal BPE configuration, presumably due to the spatial gradient in charge and in oxidation state. This work offers a paradigm for magnetoelectric actuation, that may extend the use of magneto-ionics to widespread areas beyond memories relying on electrically interconnected bits of data. This may comprise applications where electric wiring is a drawback, such as electrostimulation in medical treatments, microfluidics, magnonics or remotely actuated magnetic microrobots, amongst others. Furthermore, the actuation protocols presented here could be extrapolated to other materials, as a means to control other physical properties in a wireless manner, such as superconductivity, memristors or insulator-metal transitions, thus further spanning the range of applications and the technological impact of the obtained results.

## Methods

### Sample preparation

Room-temperature magnetron sputtering deposition of the thin films was performed in an AJA International ATC 2400 Sputtering System. For the horizontal BPE samples, 50-nm CoN films were deposited using a 50% N$_2$/50% Ar atmosphere on [100]-oriented Si substrates (8 mm × 8 mm lateral size, 0.5 mm thick), precoated with 20 nm of Ti and 60 nm of Au. The total pressure was set at a value of $3.0 \times 10^{-3}$ Torr for the sputtering. For the vertical BPE samples, a

change in lateral dimensions was adopted: layered structures of Ti (20 nm)/Au (60 nm)/Si of lateral dimension of 8 mm × 20 mm were used as the substrates for depositing the 50 nm-thick CoN films, using the same sputtering conditions as for the horizontal BPE films. The depositions were done while masking the Au layer to fix the lateral size of the magneto-ionic layers to 8 mm × 6 mm, permitting the full immersion of CoN in the electrolyte in our setup.

## Bipolar electrochemistry experiments

BPE experiments were carried out in homemade electrochemical cells, as schematically depicted in Figs. 1a and 2a. The electrolyte solution was 0.1 M KI in PC. Anhydrous PC was used a solvent to prevent CoN hydrolysis in an aqueous media, which would yield cobalt oxides. The presence of KI offers the required conductivity of the electrolyte and it also supports the complementary oxidation reaction at the driving anode and the induced anode at the bipolar electrodes. The optimal electrolyte concentrations were screened previously through a set of experiments with correlated magnetization measurements. Empirically, it was found that very small water contents improve the process, in a relatively narrow range between 200 and 800 ppm, and therefore, each experiment was performed with a parallel evaluation of the water content using the Karl Fisher method (see Section 3, Supplementary Information). The driving electrodes were a pair of parallel planar Pt plates (Goodfellow 99.95%, rectangular of 15 mm × 35 mm size) that ensured a uniform electric field across the cell. External driving voltages were applied through a power supply (Agilent B2902A) across the connected Pt electrodes immersed in the bipolar electrochemical cells. The voltage values presented throughout the manuscript always refer to external driving voltage values. After a careful optimization of distances between electrodes and sample, the Pt electrodes were distanced 20 mm from each other for the vertical BPE alignment and 25 mm for the horizontal device configuration. Thus, the applied fields correspond to $\Delta V/20$ and $\Delta V/25$ Volt/mm in vertical and horizontal configurations respectively. For the horizontal BPE devices, the samples were simply placed into a shallow pool containing a 25 mL electrolyte solution and were aligned at the centre of the Pt plates. Millimetre graded papers were put underneath the electrochemical cell to ensure the samples were positioned correctly in the centre and to ensure the reproducibility in all experiments. For the vertical BPE experiments, the samples were vertically inserted in the electrolyte with the help of home-made Teflon supports halfway between the driving Pt electrodes. In order to avoid undesired electrochemical processes and to increase the effective potential on the CoN electrode, the back of the immersed CoN sample (Pt) is electronically connected to the Au underlayer through a U-shaped Pt foil. Iodide ions form $I_2$ (or $I_3^-$) during the process, corresponding to the yellow-orange colours observed in the pictures (Sections 1 and 2, Supplementary Information). $I$ vs. $t$ curves for the connected Pt driving electrodes during the bipolar experiments are shown in the Supplementary Information, Section 9. Reversibility tests were performed applying 15 V between driving Pt electrodes for five minutes, allowing the bipolar electrode to discharge, and applying an opposite voltage of the same magnitude.

## COMSOL simulations

Electrostatic simulations of the BPE cell were carried out in each configuration with the software platform COMSOL Multiphysics, using the electrostatics module, to evaluate the initial dipole induced at the borders of immersed bipolar CoN electrode. For the horizontal configuration, it is a 3D calculation with a geometry identical to the experimental cell. The mesh size was optimized for each calculation until convergence of the physical solution of loads and potentials in the cell was achieved. The vertical configuration, on the other hand, does not allow an electrostatic 3D simulation with the real

dimensions, given the small thickness of the bipolar electrode (immersed sample) in the direction of the electric field. Therefore, an approximation was made where the immersed CoN is 1 mm in thickness is the only material taken into account. The conductivities used in the simulation were taken from the literature, i.e., $2.4 \times 10^1$ S m$^{-1}$ for CoN[30], $4.6 \times 10^{-2}$ S m$^{-1}$ for an anhydrous PC electrolyte with dissolved salts[35,36].

## Vibrating sample magnetometry

Magnetic hysteresis loops were recorded ex situ at room temperature using a VSM from Micro Sense (LOT-Quantum Design). All measurements were carried out with the applied magnetic field parallel to the film plane. The maximum magnetic field was 20 kOe. The hysteresis loops were background-corrected to subtract the linear diamagnetic/paramagnetic contributions at high fields, where ferromagnetic component signals get fully saturated. Please note that the vibrating features of the VSM and the required dimensions of samples prevent in operando conditions.

## Transmission electron microscopy

TEM and EELS measurements were performed on an FEI Tecnai G2 F20 microscope with field emission gun operating at 200 kV. Cross-sectional thin lamellae of the samples were cut by focused ion beam after the deposition of Pt protective layers and were subsequently placed onto a Cu TEM grid.

## X-ray photoelectron spectroscopy

XPS analysis were carried out using a PHI 5500 Multitechnique System spectrometer, equipped with a monochromatic Al K$_\alpha$ X-ray source (1486.6 eV) at a power of 350 W, from Physical Electronics. Background subtraction and peak fitting were performed with MagicPlot software. The XPS spectra were corrected using the C $1s$ line at 284.5 eV.

## Polarization dependent X-ray spectroscopy

Room-temperature soft XAS and XMCD measurements were carried out at the BOREAS beamline of the ALBA synchrotron (Barcelona, Spain)[37], under ultra-high vacuum conditions (base pressure below $10^{-7}$ Torr). The Co $L_{2,3}$ X-ray absorption was measured using circularly polarized light with photon helicity parallel ($\mu^+$) or antiparallel ($\mu^-$) with respect to a magnetic field of 1 Tesla, along the beam incidence direction, in a sequence of $\mu^+\mu^-\mu^-\mu^+\mu^+\mu^-\mu^-\mu^+$ to disentangle the XMCD. The average XAS and XMCD spectra were calculated as the sum and the difference, respectively, of the XAS spectra measured with two opposite helicities.

## Data availability

All data that support the findings of this study are present in the Article and/or its Supplementary Information. Source data are provided with this paper.

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

## Acknowledgements

Financial support by the European Union's Horizon 2020 Research and Innovation Programme ('BeMAGIC' European Training Network, ETN/ITN Marie Skłodowska–Curie grant N° 861145), the European Research Council (2021-ERC-Advanced 'REMINDS' Grant N° 101054687), the Spanish Government AEI (PID2020-116844RB-C21 and PDC2021-121276-C31, RTI2018-097753-B-I00, PID2021-123276OB-I00, and Severo ochoa CEX2019-000917-S), the Generalitat de Catalunya (2021-SGR-00651) and the MCIN/AEI/10.13039/501100011033 & "European Union Next-GenerationEU/PRTR" (grant CNS2022-135230) is acknowledged. The XAS/XMCD experiments were performed at BL29-BOREAS beamline at ALBA Synchrotron with the collaboration of ALBA staff. J.S. thanks the Spanish "Fábrica Nacional de Moneda y Timbre" (FNMT) for fruitful discussions. E.M. is a Serra Húnter Fellow.

## Author contributions

J.S., N.C-P., E.P. and E.M. conceived and supervised the study. Z.M. prepared the samples with input from Z.T., and L.A. helped with the fabrication of the samples with desired dimensions. Z.M., N.C-P and L.F-R. performed the bipolar electrochemistry experiments, with N.C-P conceptions. Z.M. and J.S. performed and analysed the VSM measurements. Z.M. and E.M. performed TEM and EELS measurements and analysed the results. Z.M. and E.P. performed and analysed the XPS spectroscopy. J.H-M. and Z.M. performed the XAS/XMCD experiments and analysed the data. L.F-R. and N.C-P. performed linear sweep voltammetry measurements. L.F-R. and L.A. conducted the COMSOL simulations. Z.M. wrote the manuscript with inputs from J.S., E.M., E.P., N.C-P. and L.F-R. All authors contributed to discussions and editing.

## Competing interests

The authors declare no competing interests.
