## [Peer Review File · Nature Communications]

REVIEWER COMMENTS

Reviewer #1 (Remarks to the Author):

This manuscript reports on an interesting improvement of the magneto-ionics effects by a research group extremely active in this promising field. In my opinion, the wireless actuation of the magneto-ionics effect may offer a real breakthrough in the design of devices for several applications, even if a further step as the use of a solid electrolyte is highly desirable. Nevertheless, I believe this is an important achievement in the research field and a proof of concept that deserves to be published and implemented by the scientific community.

I would like to ask the authors to comment on a couple of aspects mainly to clarify the process occurring here.

The authors observed a very different effect in the two configurations. In the horizontal one, the arising of a net magnetisation is volatile and M_s is seen to decrease with time. It has been ascribed to the charge gradient that leads to sample areas less affected. However, after 24 hrs they still observe a saturation value around 5 emu/cm³. As said, the volatile effect is ascribed to this redox/ionic lateral gradient and the authors claim that it is not observable by structural investigations. It means that the TEM investigations did not reveal any detectable effects? In my opinion the presence of well-defined hysteresis loops after 24 hrs should give rise to some magnetic phase still detectable.

The authors already report on the wired magnetoionics effects in CoN films with the very same layer stack and they ascribe the lower M_s values obtained here to be dependent on the lower voltage applied. Do they check and observe in the reported case any CoN thickness dependence on saturation magnetisation value and volatility? Especially in the horizontal configuration it may play a role together with different applied voltage values.

How about the repeatability of the measurements? Do the authors have any comments on this and on the M_s values reached during different experiments?

Concerning vertical configuration, the evolution of the magnetization process is evident. The authors comment only the M_s value as a function of voltage. However, also coercivity is seen to change together with magnetic susceptibility. Can the authors comment on this by inserting a table with those values as done in Fig. 2c and 2e for M_s ? The field scale can also be reduced to 4 kOe especially for higher applied V and time in every experiment. It has to be connected to the detected changes in the magnetic layers so nicely discussed later. In Fig. 2e where the M_s vs V behavior is reported, a slope change is clearly evident for values higher than 10 V, especially for $t = 15$ s. Do the authors have any explanation for such a slope change? It will approach a saturation value: do the authors have any idea how close it is?

I am convinced that inserting such explanation in the text, would help to clarify the experiment and mechanism.

Reviewer #2 (Remarks to the Author):

I have thoroughly reviewed the paper titled "Modulation of magnetic properties through voltage-driven ion motion and redox processes" submitted to Nature Communications. The authors propose an alternative approach to achieve voltage-driven control of magnetism through wireless bipolar electrochemistry instead of direct-contact magneto-ionics. The paper presents the results of experiments on cobalt nitride thin films immersed in electrolytes, showing tunability of magnetization and transitions between paramagnetic and ferromagnetic states. The paper is well-written and presents good quality data collected and interpreted in a careful manner. However, I have concerns about the novelty and significance of the findings.

While the use of bipolar electrochemistry is not new, the authors apply it to drive magneto-ionics processes, which is of interest. However, the absence of any new physics or chemistry involved in the study, makes it difficult to assess the significance of the reported findings. The field of magneto-ionics is moving forward at a relatively high speed and a voltage control of magnetisation

inside an electrochemical cell is not new, specially thanks to the previous work done in this same group. Nitrogen magneto-ionics has also been introduced in a previous publication by the same group.

Furthermore, the paper does not provide any evidence or discussion to support the authors' claim that the proposed approach has potential applications in bioelectronics, catalysis, neuromorphic computing, or wireless communications. The paper would benefit from a first principle demonstration of any of the different potential applications the authors claim that can be achieved using wireless electrochemistry. In addition, this method requires higher voltages than its direct wiring counterpart to achieve the same effects, therefore, it is important to demonstrate a scenario in which using this method could be of benefit despite this drawback. This would strengthen the significance of the findings and highlight the unique advantages of the proposed approach.

In conclusion, I cannot recommend this paper for publication in Nature Communications due to a lack of novelty, in particular due to the absence of new physics/chemistry or clear avenues for novel potential applications using this method.

Reviewer #3 (Remarks to the Author):

This manuscript discusses a wireless method for controlling magnetism by inducing polarization in a material immersed in an electrolyte. This method enables wireless bipolar electrochemistry, which offers an alternative approach to voltage-driven control of magnetism. The authors demonstrate the tunability of magnetization for cobalt nitride thin films, showing transitions between paramagnetic and ferromagnetic states. The effects can be either volatile or non-volatile, depending on the electrochemical cell configuration. The authors suggest that these findings could be a fundamental breakthrough that inspires future device designs for various applications, including bioelectronics, catalysis, neuromorphic computing, and wireless communications. However, the manuscript needs further attention to the following points before it can be accepted:

- 1) The switching time from paramagnetic to ferromagnetic takes a few minutes, which may not be suitable for most practical applications. The authors need to provide more evidence and discussion regarding the potential applications of this research.
- 2) The vertical bipolar electrochemistry cell needs further investigation to assess its magnetic stability and reversibility, which is very important.
- 3) The high voltage of 15 V used in the electrochemical reaction may cause other reactions on the magnetic film. The authors need to prove the stability of the electrolyte and PC at this potential to exclude other possible reactions.

Reviewer #4 (Remarks to the Author):

This work presents a new wireless architecture for magneto-ionic devices. The authors prepare a capacitive setup with an ionic liquid solution of KI + propylene carbonate. The sample is placed between the electrodes. Applying a voltage between the electrodes the authors claim that an electric double layer is generated at the surface of the sample, resulting in a large electric field at the surface which causes the magneto-ionic effect. Since the sample is freely suspended the architecture is identified as wireless.

I think the work has potential, but the authors need to provide some additional evidence about their experiment and evidence against chemical (not electrochemical) reduction. Depending on the outcome of these experiments, the results may be suitable for publication in Nature Communications.

The biggest problem that I have stems from Figure S1 and S2. These images show that the author's process prepares soluble iodine in the solution. The authors can calculate the reactivity of $2\text{CoN} + 4\text{I} \rightarrow 2\text{CoI}_2 + \text{N}_2$ and will find it to be -1.2 eV, e.g. exothermic and will occur spontaneously in the solution. This is consistent with all of the author's observations, including the visible precipitation of N_2 gas and I_2 in solution, the XPS results which showed a decreasing nitrogen

signal and the XAS results which showed Co^{2+} . As this occurs by chemical diffusion, rather than electrochemistry, it would be diffusion based, with a clear front of migration, as observed in the EELS. CoI_2 is a ferromagnetic material, with a TC of 75 K, but mixed in complex with nitrogen, it is not clear to me what the TC would be. That being said, it is also known that CoI_2 is unstable and can decompose to $\text{Co} + \text{I}_2$, which is a great magnetic material. The authors likely did not do iodine XPS (probably too high of energy) but maybe could see iodine in EELS, but iodine might leave the lattice after removing it from the solution, as I_2 .

Since this undermines the whole story, it is imperative to investigate this. I propose two follow up experiments, in one, the sample is placed at varying distances to the electrodes, which will result in a difference in iodine exposure. In the second, the electrodes are placed outside of the ionic liquid (for example, on the outside of a beaker containing the IL and sample), this will allow the electric field to penetrate without iodine formation. I also propose that the authors could submerge the sample in a solution of $\text{KI} + \text{I}_2$ (no electrodes) and see if they have a similar effect.

For the horizontal configuration, the authors should measure opposite ends of the horizontal sample and report their observation, rather than treating the sample as a whole. If the effect really is an EDL, the opposite ends of the sample will have opposite EDL polarizations, with one end encouraging N extraction (pull) and the other pushing N into the materials. If the diffusion is fast then the push end will be depleted and magnetic, and if the diffusion is slow, the pull end will be depleted and magnetic. The authors should also measure the middle of the sample, which should still be non-magnetic. If this is a chemical reaction, the end close to the anode will be magnetic, with the magnetization decreasing with distance from the electrode.

The authors make claims about the potential in the solution, but these measurements are standard for electrochemistry. The authors should use a reference electrode and measure the potential. The author's proposed mechanism requires CoN to be metallic, a point which should be made explicit in the text.

The authors should revise Figure S5, the Comsol model, as it is it took a long time to interpret. As it is in the supplement, the authors could include three panels: The electric field through the sample, through the solution, and the difference. As it is, it looks like there is a changing voltage in the metallic sample (and hence an electric field), which is not possible.

Here is a summary of the comments that I would ask to be addressed:

- (1) The authors could repeat the experiment moving the sample closer and further from the anode. If there is a uniform electric field in the solution then the voltage drop will be the same independent of the distance to the electrode. On the other hand, any chemical reaction with iodine will be strong near the anode and weak close to the cathode. This will allow the two mechanisms to be separated.
- (2) The authors could repeat the experiment with the electrodes separated from the ionic liquid, such as outside of a beaker, which will disallow any KI breakdown, but should still allow the electric field to pass through.
- (3) Use a reference electrode and quantify the electric field in the solution, then compare that electric field to the contact voltammetry curve.
- (4) Look for Iodine in the EELS results (preferably when fresh from the solution, decomposition to I_2 will result in gaseous emissions and may not be present in the sample if it sits around)
- (5) Look for changes in the sample by submerging it in a solution of $\text{KI} + \text{I}_2 + \text{PC} + \text{H}_2\text{O}$ without an electric field
- (6) In the horizontal configuration, the authors should measure opposite ends of the sample as well as the middle.

Minor Comments:

I was generally unhappy about this statement: "...subsequent structural characterization of the treated samples was hindered due to the volatile nature of the induced effects in the horizontal BPE treatments" Why was it so much more volatile compared to the vertical configuration?

The following sentence makes an important point that should be emphasized earlier, that the horizontal configuration only works near the ends of the sample:

"The volatility of the effect in the horizontal configuration was the occurrence of a redox/ionic lateral gradient within the zone where potentials are sufficiently high. Such gradient is likely to be present also for the vertical configuration (perpendicular to the sample), but with much smaller magnitude since the film is very thin and now the entire outer surface of the film (and not only the edges) contributes to the redox process."

"X-ray photoemission spectroscopy (XPS) study (Section 5, Supplementary Information) shows a decrease of the nitrogen content for the treated sample, thus corroborating the voltage-induced nitrogen ion release to the electrolyte" This just shows a decrease in valence, which is consistent with nitrogen extraction, but also iodine reaction.

The authors should report their leakage current. Also, does the leakage current change with the sample in versus not? I ask because if there is sufficient electrical conductivity, then this becomes not so much wireless, as much as current transport through the solution.

RESPONSE TO REVIEWERS' COMMENTS

Reviewer # 1

This manuscript reports on an interesting improvement of the magneto-ionics effects by a research group extremely active in this promising field. In my opinion, the wireless actuation of the magneto-ionics effect may offer a real breakthrough in the design of devices for several applications, even if a further step as the use of a solid electrolyte is highly desirable. Nevertheless, I believe this is an important achievement in the research field and a proof of concept that deserves to be published and implemented by the scientific community.

We are very pleased that Reviewer #1 considers our wireless magneto-ionic actuation an important achievement with relevant implications in the design of devices and recommends this work for publication in *Nature Communications* after a suitable revision.

I would like to ask the authors to comment on a couple of aspects mainly to clarify the process occurring here. The authors observed a very different effect in the two configurations. In the horizontal one, the arising of a net magnetisation is volatile and M_S is seen to decrease with time. It has been ascribed to the charge gradient that leads to sample areas less affected. However, after 24 hrs they still observe a saturation value around 5 emu/cm³. As said, the volatile effect is ascribed to this redox/ionic lateral gradient and the authors claim that it is not observable by structural investigations. It means that the TEM investigations did not reveal any detectable effects? In my opinion the presence of well-defined hysteresis loops after 24 hrs should give rise to some magnetic phase still detectable.

We do agree with Reviewer #1 that the hysteresis loop taken after 24 h in the horizontal configuration (Fig. 1) is consistent with a tiny fraction of a ferromagnetic counterpart, most likely a Co-rich phase (with extracted N not recovered), which has not been fully oxidized during the electrochemical relaxation process that restores electroneutrality once the external voltage is switched off. Please note that the saturation magnetization of metallic HCP-Co is around 1440 emu/cm³ at room temperature. Therefore, a value of 5 emu/cm³ indicates that we would have 0.35% of metallic Co (if metallic Co was the only ferromagnetic phase present in the sample) after 24 h. Evidence for this tiny amount of ferromagnetic phases would be very hard to find by local TEM characterization. Therefore, TEM observations on this sample were not attempted. In addition, TEM is an *ex-situ* characterization technique in which the sample (prepared in the form of a cross-sectional thin layer/lamella) is exposed to air. This would cause the natural oxidation of the remaining Co-rich phase, further hampering its characterization.

To emphasize that, in the horizontal configuration, there is a tiny permanent ferromagnetic signal, we have replaced the following sentence in page 4 "... and approaches the original untreated sample value after around 24 hours. The volatile magnetization change is most likely..." with "... M_S reduces by more than 50% within 12 hours and decreases below 10 emu cm⁻³ after around 24 hours (a value which is < 0.7% the saturation magnetization of metallic Co), evidencing that this process is virtually volatile. This magnetization depletion is most likely related to..." (page 4, line 110).

The authors already report on the wired magnetoionics effects in CoN films with the very same layer stack and they ascribe the lower M_s values obtained here to be dependent on the lower voltage applied. Do they check and observe in the reported case any CoN thickness dependence on saturation magnetisation value and volatility? Especially in the horizontal configuration it may play a role together with different applied voltage values.

We would like to thank the referee for pointing out this comment since it will help to clarify the wireless magneto-ionic approach. Along the text, voltage always refers to the external driving voltage provided by the power supply to the Pt electrodes immersed in the liquid. As illustrated in Figure 1a, the external driving voltage makes the polar propylene carbonate molecules reorient, and cations (K^+) and anions (I^-) diffuse towards opposite directions. While no electric field exists within a bulk metal, the surface where the conducting material ends undergoes a polarization that distorts the ionic electrolyte distribution, generating an induced potential difference between the CoN and the electrolyte, which is lower in magnitude than the external driving voltage. To emphasize throughout the manuscript that the given voltage values always refer to external driving voltage, several small changes have been made in the revised manuscript: “Voltage” has been replaced with “external driving voltage” in Figure 2e; in line 284 of the “Methods” section, the following sentence “Driving voltages were applied through an external power supply (Agilent B2902A) across the connected Pt electrodes immersed in the bipolar electrochemical cells.” has been replaced with “External driving voltages were applied through a power supply (Agilent B2902A) across the connected Pt electrodes immersed in the bipolar electrochemical cells. The voltage values presented throughout the manuscript always refer to external driving voltage values”; caption of Figure S4 “**Fig. S4** Simulation of the interfacial potential difference profile (induced voltage negative minus the electrolyte voltage) through the cell (the sample is positioned in the centre) for CoN in both horizontal (purple line) and vertical (orange line) configurations. Please note the different scales in the induced voltage.” has been rewritten as “**Fig. S5** COMSOL simulation of the potential distribution with respect to the electrolyte voltage when applying an external driving voltage of 15 V for the horizontal (a) and vertical (b) configurations. The same scale has been used for easy comparison. The red dashed vertical lines indicate the sample boundaries. Please note that the nm scale of the vertical configuration has been simplified to mm, since COMSOL cannot achieve such relative scales difference”.

As stated by the referee, the interplay between CoN thickness and employed configuration is likely to play a role in the magneto-ionic response. If the effect is maintained without additional variables, probably, when increasing the thickness of the CoN layer, non-volatility might be hampered in the vertical configuration since larger redox gradients will be generated across the CoN layer. Conversely, in the horizontal configuration, even though the induced magneto-ionic changes may remain volatile, they might become more pronounced since a larger volume of material will be polarized. This is worth being investigated and we plan to do an exhaustive study in the future. However, we have observed before that most systems are more complex than expected when using the bipolar configurations. Specially, the potential gradients across the sample, and the redox reactions related to them, may cause resistance changes that also modify the induced polarization. Those resistance changes depend on the thickness of the material and the extent at which the reaction penetrates within the material. Therefore, the dependence of the effects with the thickness is expected to be rather complex. As an example, recently, some of the authors of the current manuscript have tackled the changes in conductivity that may appear during bipolar treatment (which modify the material in its space geometry) in Cu-CuO systems, and have studied how these changes modify the induced potentials, resulting in zones of zero charge and zones with alternance of induced dipoles of different signs, yielding fringes of different materials formed in the direction perpendicular to

the applied field, with spatial and time oscillatory behaviors (see: Fuentes-Rodríguez, L. et al., *Mater. Chem. Front.* **6**, 2284 (2022)). The system we report here, with 50 nm in thickness, is still holding the underlying conductivity of Au, and therefore, those effects are negligible. However, raising the thickness will almost certainly add a number of variables, such as conductivity, change of dipoles orientation, and space and time evolution of the process. Therefore, the study of the bipolar effect in thicker coatings requires a deep analysis of the local resistance changes and local resolution spectroscopic techniques. These are certainly worthy experiments to be undertaken but will require extensive work that is beyond the scope of the current manuscript. Nevertheless, we agree with the referee that the thickness-dependent study constitutes an interesting direction to be explored in the near future.

How about the repeatability of the measurements? Do the authors have any comments on this and on the M_S values reached during different experiments?

We would like to highlight that the measurements in both configurations are reproducible. The results have been repeated several times, resulting always in the same trends. The time evolution of the induced effects in the horizontal configuration is reproducible. The reproducibility is also always good in vertical configuration, at driving potentials below 15 V. Once solvent reactions start to occur (i.e., for higher driving voltages), then gas is formed and the induced magneto-ionic effects worsen, as explained in the manuscript.

Concerning vertical configuration, the evolution of the magnetization process is evident. The authors comment only the M_S value as a function of voltage. However, also coercivity is seen to change together with magnetic susceptibility. Can the authors comment on this by inserting a table with those value as done in Fig. 2c and 2e for M_S ? The field scale can also be reduced to 4 kOe especially for higher applied V and time in every experiments. It has to be connected to the detected changes in the magnetic layers so nicely discussed later. In Fig. 2e where the M_S vs V behavior is reported, a slope change is clearly evident for values higher than 10 V, especially for $t = 15$ s. Do the authors have any explanation for such a slope change? It will approach a saturation values: do the authors have any idea how close it is? I am convinced that inserting such explanation in the text, would help to clarify the experiment and mechanism.

The reviewer is right about the added value of commenting on the variation of coercivity (H_C) and squareness (ratio between remanent saturation M_R and saturation magnetization, M_R/M_S (%)) with time and driving voltage in the vertical configuration (see Table below, which has been added as Table S1 in the Supplementary Information). However, since coercivity (H_C) and remanent saturation M_R are parameters which are dependent on the microstructure (i.e., extrinsic) and the evolution of microstructure can be rather complex, the interpretation of the voltage/time dependence of H_C and M_R/M_S might be not straightforward. Anyhow, the following sentence has been added in the main text (page 7, line 159) to account for the main trends on how H_C and M_R/M_S evolve with voltage and time “With time and voltage, the variation of coercivity (H_C) shows no clear trends (values ranging from around 50 to 200 Oe), whereas the evolution of squareness (ratio between remanent saturation M_R and saturation magnetization M_S : M_R/M_S (%)) tends to increase with both time and voltage (going from 7 to 66 %), as shown in Table S1 in the Supplementary Information. This is consistent with an increased amount of ferromagnetic counterpart (i.e., the evolution with time and voltage of M_S), with a better defined in-plane magnetic anisotropy.”.

Table S1. Dependence of the coercivity and the squareness ratio (M_R/M_S) on the driving voltage and actuation time.

	H_C (Oe)	M_R/M_S (%)
15V 2.5min	120	17.3
15V 5min	144	35.4
15V 10min	148	51.1
15V 15min	74	27.2
15V 20min	197	58.5
5V 5min	56	6.7
10V 5min	66	7.6
15V 5min	144	35.1
20V 5min	106	64.3
5V 15min	109	12.0
10V 15min	122	34.5
15V 15min	74	26.4
20V 15min	113	66.1

Following the reviewer’s suggestion, in Figures 2b and 2d, the applied field scale has been reduced to ± 4 kOe to better visualize the shape of the hysteresis loops.

To initiate voltage-driven ion motion and, thus, magneto-ionics, a voltage above a certain threshold, often called onset voltage, is required. When dealing with highly nanostructured phases, as it is the case of the investigated CoN films, a distribution of onset voltages is expected. Therefore, the slope change between 10 and 15 V in Figure 2e evidences that 15 V is above the onset voltage for magneto-ionics in this system. To comment on this slope change, the following sentence has been added in the main text when discussing Figure 2e in page 7, line 168: “Between 10 and 15 V, a pronounced slope change occurs, evidencing that 15 V is above the onset voltage for magneto-ionics in this system.”.

Magneto-ionic effects are usually slow and, in the employed approach, a time framework of minutes is required to see significant changes. Thus, in Fig. 2e, the values are likely not far from saturation. We do believe that potentials higher than 10 V mark the start of a counter electrode reaction. That is, above such potential, the oxidation of I^- to I_2 may occur, facilitating also the reduction of CoN and affecting the slope of the curve.

Reviewer #2

I have thoroughly reviewed the paper titled "Modulation of magnetic properties through voltage-driven ion motion and redox processes" submitted to Nature Communications. The authors propose an alternative approach to achieve voltage-driven control of magnetism through wireless bipolar electrochemistry instead of direct-contact magneto-ionics. The paper presents the results of experiments on cobalt nitride thin films immersed in electrolytes, showing tunability of magnetization and transitions between paramagnetic and ferromagnetic states. The paper is well-written and presents good quality data collected and interpreted in a careful manner. However, I have concerns about the novelty and significance of the findings. While the use of bipolar electrochemistry is not new, the authors apply it to drive magneto-ionics processes, which is of interest. However, the absence of any new physics or chemistry involved in the study, makes it difficult to assess the significance of the reported findings. The field of magneto-ionics is moving forward at a relatively high speed and a voltage control of magnetisation inside an electrochemical cell is not new, specially thanks to the previous work done in this same group. Nitrogen magneto-ionics has also been introduced in a previous publication by the same group.

The reviewer recognizes that the work presented in this manuscript merges two important fields in materials science, (i) magneto-ionics and (ii) bipolar electrochemistry, both of them appealing on their own and with rising scientific interest. Importantly, such merging, shown here for the first time, allows to control magnetism with voltage in a wireless manner. We believe that this, by itself, opens new avenues in the research field of magnetoelectricity, bringing new prospects from both fundamental and applied points of view, thus making our work of sufficiently general scientific interest to be published in Nature Communications.

The phenomenon of bipolar electrochemistry is known since a few decades ago and it has been found successful by the community for the synthesis of nanostructures and polymeric films, analytical chemistry, micro-object propulsion or water splitting (see for example: Mavr , F. et al., *Anal. Chem.* **82** 8766 (2010); Loget, G. et al., *Anal. Bioanal. Chem.* **400** 1691 (2011); Fleischmann M. et al., *J. Phys. Chem.* **90** 6392 (1986)). Yet, such phenomenon has never been used before for the control of magnetic properties of materials in a wireless manner. In this new research direction, there are still many factors to be elucidated, which can lead to new processes and applications, still hard to be envisaged, as described in the manuscript. So far, the main ways to manipulate magnetism in a wireless manner have been through ion irradiation or by illuminating with high-energy light. Both strategies have raised significant attention in the community and have resulted in impactful publications (e.g., Chappert, C. et al., *Science* **280** 1919 (1998); Siegrist, F. et al., *Nature* **571**, 240 (2019); Afanasiev, D. et al., *Nat. Mater.* **20**, 607 (2021)). Our new approach for the wireless control magnetism could be similarly impactful in the future and it will, for sure, be the starting point of forthcoming new studies.

We certainly do not agree with the vision that there is absence of any new physics or chemistry involved in the study. Our previous works on magneto-ionics using voltage-driven nitrogen ion migration (always by directly wiring the sample to a power supply) mainly resulted in non-volatile (i.e., permanent) effects. Here, we observe volatile effects using a horizontal bipolar electrochemistry configuration and we explain this volatility as being due to the strong concentration gradients of the involved chemical species. Also, we perform a detailed characterization of the involved electrochemical processes, which are correlated with the underlying physical aspects governing non-wired magneto-ionics. The possibility of inducing physical changes without direct wiring is significant in biosystems or in electronics, and in the fundamental understanding of the nature of all magneto-ionic processes in general, as summarized in the manuscript.

Furthermore, the paper does not provide any evidence or discussion to support the authors' claim that the proposed approach has potential applications in bioelectronics, catalysis, neuromorphic computing, or wireless communications. The paper would benefit from a first principle demonstration of any of the different potential applications the authors claim that can be achieved using wireless electrochemistry. In addition, this method requires higher voltages than its direct wiring counterpart to achieve the same effects, therefore, it is important to demonstrate a scenario in which using this method could be of benefit despite this drawback. This would strengthen the significance of the findings and highlight the unique advantages of the proposed approach. In conclusion, I cannot recommend this paper for publication in Nature Communications due to a lack of novelty, in particular due to the absence of new physics/chemistry or clear avenues for novel potential applications using this method.

We certainly believe that the results of this manuscript could be the starting point for the development of new applications based on the wireless control of magnetism or electronics. As an example, magnetoelectric stimulation without directly wired electrodes could allow new clinical processes. Namely, we could imagine stimulating the retina through these induction processes, or generating ferromagnetic properties to materials inserted in cells (e.g., magnetotherapy based on the use of wireless voltage-induced ferromagnetism in nanoparticles). Note that the applied driving voltages used here are comparable or even lower than those utilized to induced nitrogen magneto-ionic effects in wired CoN samples (De Rojas, J. et al., *Nat. Commun.* **11** 5871 (2020)). This has been achieved by properly engineering the electrolytic bath by incorporation of KI in suitable concentrations.

The obtained results have potential for emerging neuromorphic computing applications. We have recently reported on the use of magneto-ionics to build voltage-controlled artificial synapses (Tan Z. et al., *Mater. Horiz.* **10**, 88 (2023)). Some of the basic synaptic functionalities (such as analog control of magnetization for multi-level memories, voltage-amplitude dependent plasticity, time-dependent plasticity or voltage-controlled potentiation/depression of the synaptic signal), are also demonstrated in the current manuscript (e.g., Figs. 1 and 2) and can be achieved in a wireless manner.

Voltage actuation (using conventionally wired samples) has been also reported as a suitable strategy to control spin waves in a variety of magnetic materials (see e.g., Zhao, S. et al., *Adv. Electron. Mater.* **6**, 1900859 (2020); Rana B. et al., *Commun. Phys.* **2**, 90 (2019)). Voltage control of the propagation of spin waves can lead to energy efficient magnonic logic and communication devices. In this sense, being able to induce such effects without directly wiring onto the actuated samples could lead to new strategies to modulate wireless transfer of magnetic/spintronic information.

Finally, it has been reported that some catalytic and biocatalytic reactions (such as hydrogen evolution reaction) can be boosted with the use of magnetically-functionalized electrodes (Willner I. et al., *Angew. Chem. Int. Ed.* **42**, 4576 (2003); Jiang X. et al., *ChemSusChem* **15**, e202201551 (2022)). Hence, remote manipulation of the magnetic properties of catalysts through a wireless driving voltage could be of interest for magneto-catalysis, for example.

Direct demonstration of these applications is beyond the scope of the current manuscript. Yet, given the above descriptions, it is clear that our results have the potential to impact the areas of bioelectronics, catalysis, neuromorphic computing, or wireless communications. What is more, wireless magneto-ionics could be the most effective way to actuate on miniaturized magnetic devices with voltage, when their size becomes so small that electric contacts can affect the induced changes in magnetic properties. We expect that the seminal results presented here will be a source of inspiration for other groups to pursue further research on this topic and implement the above (and other) technological applications. As it always happens,

reaching the final applications takes time, and technical developments are always preceded by more basic works like the one we are reporting here.

Reviewer #3

This manuscript discusses a wireless method for controlling magnetism by inducing polarization in a material immersed in an electrolyte. This method enables wireless bipolar electrochemistry, which offers an alternative approach to voltage-driven control of magnetism. The authors demonstrate the tunability of magnetization for cobalt nitride thin films, showing transitions between paramagnetic and ferromagnetic states. The effects can be either volatile or non-volatile, depending on the electrochemical cell configuration. The authors suggest that these findings could be a fundamental breakthrough that inspires future device designs for various applications, including bioelectronics, catalysis, neuromorphic computing, and wireless communications. However, the manuscript needs further attention to the following points before it can be accepted:

1) The switching time from paramagnetic to ferromagnetic takes a few minutes, which may not be suitable for most practical applications. The authors need to provide more evidence and discussion regarding the potential applications of this research.

We thank the referee for pointing this out. We acknowledge that the reported magnetic switching time (a few minutes) may need further improvement for practical devices working at high frequencies. Still, the speed limit of property control via electrochemistry may be of less relevance for certain application domains, such as electrostimulation, magnetic therapies or sensing.

In terms of applications impact, the experimental scheme demonstrated in our wireless magneto-ionic device, based on bipolar electrochemistry, may inspire other research into wireless voltage control of physical/chemical properties of materials such as superconductivity or metal-insulator transitions, and it is likely to open new avenues in iontronics and wireless magnetoelectric devices, as described in the original Main Text. This work may also inspire other application domains on the wireless control of magnetism. As an example, magnetoelectric stimulation without directly wired electrodes could allow new clinical processes. Namely, we could imagine stimulating the retina through these induction processes, or generating ferromagnetic properties to materials inserted in cells (e.g., magnetotherapy based on the use of wireless voltage-induced ferromagnetism in nanoparticles). Note that the applied driving voltages used here are comparable or even lower than those utilized to induced nitrogen magneto-ionic effects in wired CoN samples (De Rojas, J. et al., *Nat. Commun.* **11** 5871 (2020)). This has been achieved by properly engineering the electrolytic bath by incorporation of KI in suitable concentrations.

The obtained results have potential for emerging neuromorphic computing applications. We have recently reported on the use of magneto-ionics to build voltage-controlled artificial synapses (Tan Z. et al., *Mater. Horiz.* **10**, 88 (2023)). Some of the basic synaptic functionalities (such as analog control of magnetization for multi-level memories, voltage-amplitude dependent plasticity, time-dependent plasticity or voltage-controlled potentiation/depression of the synaptic signal), are also demonstrated in the current manuscript (e.g., Figs. 1 and 2) and can be achieved in a wireless manner.

Voltage actuation (using conventionally wired samples) has been also reported as a suitable strategy to control spin waves in a variety of magnetic materials (see e.g., Zhao, S. et al., *Adv. Electron. Mater.* **6**,

1900859 (2020); Rana B. et al., *Commun. Phys.* **2**, 90 (2019)). Voltage control of the propagation of spin waves can lead to energy efficient magnonic logic and communication devices. In this sense, being able to induce such effects without directly wiring onto the actuated samples could lead to new strategies to modulate wireless transfer of magnetic/spintronic information.

Finally, it has been reported that some catalytic and biocatalytic reactions (such as hydrogen evolution reaction) can be boosted with the use of magnetically-functionalized electrodes (Willner I., *Angew. Chem. Int. Ed.* **42**, 4576 (2003); Jiang X. et al., *ChemSusChem* **15**, e202201551 (2022)). Hence, remote manipulation of the magnetic properties of catalysts through a wireless driving voltage could be of interest for magneto-catalysis, for example.

Direct demonstration of these applications is beyond the scope of the current manuscript. Yet, given the above descriptions, it is clear that our results have the potential to impact the areas of bioelectronics, catalysis, neuromorphic computing, or wireless communications. In most of the above examples, the actuation time is not so critical and if one could reduce it to be of a few seconds or less, the wireless magneto-ionic approach could already show its potential. Importantly, the switching time could be potentially improved by tuning several experimental parameters, for example, changing to smaller ions (e.g., Li^+ or H^-) or different electrolytes, reducing the thickness of the magneto-ionic layer or decreasing the distance between the Pt driving electrodes. For example, by reducing the thickness of the magneto-ionic material from 200 nm to 5 nm, an around 7.4-fold enhancement of ion motion rate (as large as $5.9 \text{ emu cm}^{-3} \text{ s}^{-1}$) is achieved in directly wired CoN [see Z. Tan et al, *Mater. Horiz.* **10**, 88 (2023)]. In the same paper, magneto-ionic processes generated during voltage actuation with pulses of 10 ms were observed. The authors believe that exploring fast wireless magnetic switching by tuning the above-mentioned parameters is particularly appealing for future research directions. Finally, we would like to point out that wireless magneto-ionics could be the most effective way to actuate on miniaturized magnetic devices with voltage, when their size becomes so small that electric contacts can affect the induced changes in magnetic properties.

We expect that the seminal results presented here will be a source of inspiration for other groups to pursue further research on this topic and implement the above (and other) technological applications. As it always happens, reaching the final applications takes time, and technical developments are always preceded by more basic works like the one we are reporting here.

2) The vertical bipolar electrochemistry cell needs further investigation to assess its magnetic stability and reversibility, which is very important.

Thank you for this valuable suggestion. The vertical configuration is highly reproducible in terms of the generated M_S versus driving voltage and its stability, and all experiments have been repeated several times with consistent results. Magnetic reversibility tests have been done now in this configuration (see the newly included Fig. S7, Section 7, Supplementary Information). For this purpose, a CoN film was actuated at an external driving voltage of 15 V for 5 min, giving rise to a M_S generation of 47.2 emu cm^{-3} (in red). Upon discharging the bipolar electrode and then reversing the driving voltage to -15 V , the M_S decreased considerably to 5.7 emu cm^{-3} (in blue), representing an 88 % drop in M_S . The remanent magnetic signal observed here may correspond to the extent in which nitrogen ions initially released to the solution are not able to be reinserted back to the sample. The chemistry and kinetics associated with this process needs a further development, but it is important to point out that the wireless magnetoionic effect can be made largely reversible.

Fig S7 Room-temperature hysteresis loops for the as-grown (in black), treated (in red) and recovered (in blue) samples. The actuation is conducted under a condition of 15 V / 5 min, and the recovery is done by firstly discharging and then applying -15 V / 5 min (reversing the polarity of driving voltage). Both are in vertical BPE configuration.

To emphasize this point, the following sentences have been added in the revised version of the main text:
(page 7, line 183):

“Finally, the reversibility characterization of the magnetic change (Section 7, Supplementary Information) suggests the existence of quasi-reversibility for short treatment times (5 min), evidencing that a large fraction of the sample recovers within such time scale, possibly through redistribution of the remaining N ions.”

(page 15, line 301):

“Reversibility tests were performed applying 15 V between driving Pt electrodes for five minutes, allowing the bipolar electrode to discharge, and applying an opposite voltage of the same magnitude.”

3) The high voltage of 15 V used in the electrochemical reaction may cause other reactions on the magnetic film. The authors need to prove the stability of the electrolyte and PC at this potential to exclude other possible reactions.

We appreciate the concerns of Referee #3 regarding the high voltages used and the electrolyte stability under such conditions. Starting with the high voltage issue, 15 V is actually lower than the voltage values used to induce magneto-ionic effects (through direct wiring to the sample) in similar nitride films (see, e.g., De Rojas, J. et al., *ACS Appl. Mater. Interfaces* 13, 30826 (2021); Jensen, C.J. et al., *ACS Nano* 17, 6745 (2023)). Importantly, the authors would like to clarify that, throughout the manuscript, the applied voltages always refer to the external driving voltage provided by the power supply to the Pt driving electrodes. As

illustrated in Figure 1a, the external driving voltage makes the polar propylene carbonate molecules reorient, and cations (K^+) and anions (I^-) diffuse along opposite directions. While no electric field exists within a 'bulk' metal, electric polarization is induced at the surface where the conducting material ends and this distorts the ionic electrolyte distribution, generating an induced potential difference between the CoN and the electrolyte, which is much lower in magnitude than the external driving voltage. This is evidenced from our COMSOL simulations (see Figure S5), where the induced potential difference at the poles are calculated to be 7.2 V and 1.0 V when an external driving voltage of 15 V is applied for the horizontal and vertical configurations, respectively. To emphasize that the given voltage values always refer to external driving voltage, "Voltage" has been replaced with "external driving voltage" in Figure 2e. In addition, in line 284 of the "Methods" section, the following sentence "Driving voltages were applied through an external power supply (Agilent B2902A) across the connected Pt electrodes immersed in the bipolar electrochemical cells." has been replaced with "External driving voltages were applied through a power supply (Agilent B2902A) across the connected Pt electrodes immersed in the bipolar electrochemical cells. The voltage values presented throughout the manuscript always refer to external driving voltage values".

Moving on to the issue of electrolyte stability, at or below 15 V driving voltage, all evidence says that no other secondary reactions occur at the CoN with the exception of the sister-reaction I^- oxidation to I_2 at the opposite positive pole. This particular secondary reaction is restricted to the electrolyte at the bipolar induced anode, and facilitates the reduction of CoN, making the magneto-ionic change possible at much lower voltage than for direct wired experiments. There is threshold at which other reactions from the solvent occur, as explained in the text, and it corresponds to 20 V external voltage.

We have clarified both points in page 6, line 141 of the revised manuscript. "In turn, the electrolyte surrounding the positive pole also undergoes a chemical reaction, where I^- is oxidized at the induced anode, forming I_2 or I_3^- , leading to the observed orange colour in the electrolyte (Section 2, Supplementary Information). Simultaneous secondary reactions are possible in both poles at the largest applied potentials. For example, under the application of 15 V for more than 5 min, it is observed that bubbles start to appear on the induced cathode of the bipolar electrode surface, which could be related to the formation of N_2 gas after reduction of CoN, or to the complex reduction of the propylene carbonate electrolyte to aliphatic species²⁶. Since the connected Pt cathode also shows gas evolution the most probable reaction is derived from the solvent".

Reviewer #4

This work presents a new wireless architecture for magneto-ionic devices. The authors prepare a capacitive setup with an ionic liquid solution of KI + propylene carbonate. The sample is placed between the electrodes. Applying a voltage between the electrodes the authors claim that an electric double layer is generated at the surface of the sample, resulting in a large electric field at the surface which causes the magneto-ionic effect. Since the sample is freely suspended the architecture is identified as wireless. I think the work has potential, but the authors need to provide some additional evidence about their experiment and evidence against chemical (not electrochemical) reduction. Depending on the outcome of these experiments, the results may be suitable for publication in Nature Communications.

The biggest problem that I have stems from Figure S1 and S2. These images show that the author's process prepares soluble iodine in the solution. The authors can calculate the reactivity of $2\text{CoN} + 4\text{I}^- \rightarrow 2\text{CoI}_2 + \text{N}_2$ and will find it to be -1.2 eV, e.g. exothermic and will occur spontaneously in the solution. This is consistent with all of the author's observations, including the visible precipitation of N_2 gas and I_2 in solution, the XPS results which showed a decreasing nitrogen signal and the XAS results which showed Co^{2+} .

We appreciate the referee's suggestion. Despite that calculation, there is neither a spontaneous reaction between CoN and iodide, nor between CoN and iodine. As shown now in a representative spectrum (Figure S8a), no iodine signal is observed in the XPS survey spectra of treated samples. Moreover, we do not detect any traces of iodine from our energy dispersive X-ray spectroscopy (EDX) experiments (see a representative spectrum below).

Also, a blank sample, simply immersed in the electrolyte media (with no electric field applied externally), has been prepared, and no evidence of CoI_2 formation is observed by XPS. No M_s increase is observed either (Fig. S4).

Therefore, the process taking place does not involve a direct reaction as the referee suggests from exothermic calculations, and requires the application of an electric field to occur. We feel confident that the referee will see clearly from a blank immersed sample, that no direct chemical reaction occurs, on the basis of neither change in M_s , nor detection of I signal by XPS. All the observations including I_2 formation, occur only upon application of the electric field, and in samples subject to the electric field, but never in absence of it. Therefore, no direct (chemical) reaction occurs.

Fig. S8 a, General survey spectra for the as-grown sample and treated sample at a driving voltage of 15 V for 15 min.

Fig. (only in the response letter): EDX spectrum of the sample treated under 15 V / 15 min in vertical BPE configuration.

As this occurs by chemical diffusion, rather than electrochemistry, it would be diffusion based, with a clear front of migration, as observed in the EELS.

Electrochemistry always occurs at the interface between an electrode and an electrolyte and is always accompanied by chemical diffusion, of course. However, it is worth recalling that it is initiated by the existence of a charged interface. Such interface may be directly connected to the circuit like in standard electrochemistry, or through the electrolyte, without direct wiring. In this case, the induction of poles at the borders of the material, with opposite sign, comes along with a voltage profile across the sample and in the direction of the electric field. This has been shown clearly in other publications and has also been simulated by electrostatic methods (see: Fuentes-Rodríguez, L. et al., *J. Electrochem. Soc.*, **169**, 016508 (2022); Fuentes-Rodríguez, L et al., *J. Phys. Chem. C* **125**, 16629 (2021); Fuentes-Rodríguez, L. et al., *Mater. Chem. Front.* **6**, 2284 (2022)).

We use the word unwired, instead of “unconnected”, because evidently there is electric contact through the ionic electrolyte. In fact, the process cannot exist without the electrolyte. The actual potential at the poles deviates from the externally applied value because of the electrolyte voltage drop, in addition, as it has been explained and developed in more detail in some of our previous works (see: Fuentes-Rodríguez, L. et al., *J. Electrochem. Soc.*, **169**, 016508 (2022); Fuentes-Rodríguez, L. et al., *J. Electrochem. Soc.*, **169**, 016508 (2022)).

We have included now a new plot (panel b in Figure 1) to clarify the induction of a dipole vs. the electrolyte.

CoI₂ is a ferromagnetic material, with a T_C of 75 K, but mixed in complex with nitrogen, it is not clear to me what the T_C would be. That being said, it is also known that CoI₂ is unstable and can decompose to Co+I₂, which is a great magnetic material. The authors likely did not do iodine XPS (probably too high of

energy) but maybe could see iodine in EELS, but iodine might leave the lattice after removing it from the solution, as I₂.

As mentioned above, a XPS survey spectra was collected, in which it is clear that iodine is not present (signals near 630 to 615 eV are absent). This had not been included in the original version of the article, and we have added this result now in the supplementary material file (Fig. S8a). Thus, neither immersion nor the electrochemical process induce inclusion of iodine in the sample.

We would also like to remind the referee that the M_S magnetic data are registered at room temperature. CoI₂ being ferromagnetic at much lower temperatures would not contribute to a room temperature ferromagnetic signal. Indeed, it might be possible that a new phase containing N and I could be created, but there is no evidence of iodine in the blank sample or any other electric field sample experiments, therefore this possibility is unlikely to occur.

Since this undermines the whole story, it is imperative to investigate this. I propose two follow up experiments, in one, the sample is placed at varying distances to the electrodes, which will result in a difference in iodine exposure.

We sincerely believe that the discussion benefits the article, but significantly gives more value to our observations, rather than undermining them, since we can clarify now that no spontaneous reaction aside from electrochemistry occurs. We also appreciate the suggestion of complementary experiments to provide clear-cut evidence of the mechanism taking place in our set-up.

However, we must remind here that, as mentioned in the manuscript, we have given the optimized field-distance setting. Before that, several distances and voltages were screened (of course, the larger the distance, the smaller the external field for the same voltage). The same effect was observed for the same resulting electric fields, even if iodine gas produced at the Pt driving electrode was far. Again, XPS analysis did not show any I signal in the corresponding spectra of the sample.

We want to clarify here that the induced cathode and anode at the CoN horizontal sample come always simultaneously, as expected. CoN reduces at the anode, but there is always a reaction at the anode, necessarily, in this case I₂ formation. Even if we place the Pt further away, I₂ will form at the CoN induced anode surface, and when that would occur, it would not modify the material, since it would occur only in the solution, without reacting with the material.

We have clarified that point in the revised version of the manuscript (Page 3, line 92).

“Note that neither visible chemical reactions nor magnetic properties changes are observed upon immersion of the CoN in the electrolyte with the absence of external applied voltages (Section 4, Supplementary Information).”

In the second, the electrodes are placed outside of the ionic liquid (for example, on the outside of a beaker containing the IL and sample), this will allow the electric field to penetrate without iodine formation. I also propose that the authors could submerge the sample in a solution of KI+I₂ (no electrodes) and see if they have a similar effect.

Indeed, the process is based on the existence of an electrolyte, that is, ionic conductivity is a must. Furthermore, the actual polarization inducing an electrochemical reaction at the interface means that the

CoN material has to be in intimate contact with the electrolyte, in other words, it is at an interface with the electrolyte. The process cannot function without the electrolyte.

We are not sure if we understand the referee's comment correctly: External driving electrodes cannot be outside an electrolyte media, as the referee surely knows. If the referee means using a salt bridge for the connected electrodes, so that the electrolyte near the driving Pt electrodes does not contaminate the electrolyte where CoN is immersed, we want to emphasize that we do not avoid the presence of I_2 because I_2 also forms at the opposite pole of the reduction of the sample, CoN. Again, we must recall that two reactions are required at the opposite poles, and therefore the iodine reaction makes possible the reduction of CoN and the global process. Other examples do exist in literature, where the complementary reaction facilitates the reaction of interest. In all cases, as here, a blank sample is measured, and possible interactions have been checked (see, for example, Fuentes-Rodriguez, L et al., *J. Phys. Chem. C* **125**, 16629 (2021)).

We have also immersed the CoN sample in the solution without externally applying voltage, as mentioned above. No visible reactions took place. As seen in Fig. S4 in Section 4, Supplementary Information, magnetometry measurements shows no changes in magnetic moment compared to that of the as-grown sample. We have clarified that point in page 3, line 92, as mentioned above.

Fig. S4 Room-temperature hysteresis loops for the as-grown CoN films with (in red) or without (in black) submerging in the liquid electrolyte. The submersion duration is 15 minutes. Note the low magnetic moment signal in the Y-axis for both cases.

For the horizontal configuration, the authors should measure opposite ends of the horizontal sample and report their observation, rather than treating the sample as a whole. If the effect really is an EDL, the opposite ends of the sample will have opposite EDL polarizations, with one end encouraging N extraction (pull) and the other pushing N into the materials.

We thank the reviewer for this insightful comment. Certainly, opposite poles have opposite charges, becoming an anode and cathode respectively. However, the processes do not need to be a mirror of each other. This has been repeatedly reported in many publications, including ours. An example is the reduction

of IrO_x which is accompanied by the water oxidation in the opposite pole (Fuentes-Rodriguez, L et al., *J. Phys. Chem. C* **125**, 16629 (2021)).

In our case, N may move out of the CoN material, given its ionic-mixed conductivity, at the induced cathode. Therefore, changes in magnetic properties are observed. But if the electrolyte does not contain N ions as a source, their intercalation in the sample in the opposite pole is not possible. Instead, another electrochemical reaction takes place, namely, the Γ^- oxidation to I_2 , as we have chosen KI reagent to be the anode contributor.

All data reported in the manuscript confirms the occurrence of the following reactions: reduction of CoN at the induced cathode, and oxidation of Γ^- to I_2 in the induced anode nearby solution.

A new experiment has been carried out in which the horizontal treated sample has been split in three parts, as the referee suggests, corresponding to the negative pole, the center of the sample and the positive pole. Indeed, a different M_S is found for each part, as represented now in newly added Fig. S6, Section 6, Supplementary Information. A gradient in M_S does exist, i.e., M_S decreases from 50 emu cm^{-3} (in red) for the negative pole to 30 emu cm^{-3} (in blue) for the sample center and finally to around 18 emu cm^{-3} (in green) for the positive pole. The largest M_S value occurs at the negative pole, as expected. It is necessary to note here that since the gradient evolution is dynamic, and the measurements are sequential, the absolute magnitudes may vary to some extent depending on the order the hysteresis loops are measured. But, still, the gradient remains after the first hour, as expected from previous observations (see Fig. 1c in the revised manuscript). Such experiment confirms our hypothesis of the redox gradient existing in the horizontal configuration sample and discards any possibility of a direct chemical reaction.

Fig. S6 *Ex situ* consecutive measurements of hysteresis loops corresponding to the negative and positive poles and the center of the sample, evidencing the redox gradient achieved after treatment. Note that recording each loop takes about 20 min. The figure on the left shows a schematic illustration of the sample in the cell.

If the diffusion is fast then the push end will be depleted and magnetic, and if the diffusion is slow, the pull end will be depleted and magnetic. The authors should also measure the middle of the sample, which should

still be non-magnetic. If this is a chemical reaction, the end close to the anode will be magnetic, with the magnetization decreasing with distance from the electrode.

We agree with the referee's statement on the ideal situation; however, the system is dynamic. It is certainly desirable to measure an intensive property at specific coordinates, and, if possible, *in situ*, as we have done before in the references cited. Measuring an extensive property like M_S , however, is not the same. In the revised version of the paper, we report results on a treated sample which has been cut in three parts, and measured each of them independently. A clear difference in M_S is observed, with the M_S being the largest in the sample part corresponding to the negative pole.

Since the gradient is not static but dynamic in the horizontal configuration, as it has been observed (the driving force being the gradient itself), the extensive measurements could never be quantitative either. Despite this fact, within the first two hours, the M_S signal corresponding to the opposite pole is larger than the blank, in agreement with the relaxation model we propose.

Gradients and clarifications are discussed now in page 4, line 115: The hysteresis loop measurements on the individual negative and positive poles and the central part, split from a sample after the bipolar treatment, confirm a significant gradient in the corresponding M_S with larger M_S enhancement towards the positive pole (Section 6, Supplementary Information). On the basis of these results, we...

The authors make claims about the potential in the solution, but these measurements are standard for electrochemistry. The authors should use a reference electrode and measure the potential.

We are thankful to the referee for this comment. It is certainly a straightforward intuitive idea, however, it does not work with induced dipoles.

The authors are aware that measuring the potential at the electrode of interest is standard for "connected" electrodes, and reference electrodes are indeed used in many electrochemical-based techniques. However, the induced dipole at the immersed CoN or any other immersed conducting material, discharges as the poles are connected through a circuit (Fuentes-Rodríguez, L. et al., *J. Electrochem. Soc.*, **169**, 016508 (2022)). An approximation may be made described in reference JES, and also simulations based on electrostatic models have been rather successful (Fuentes-Rodríguez, L. et al., *J. Electrochem. Soc.*, **169**, 016508 (2022); Fuentes-Rodríguez, L. et al., *J. Phys. Chem. C* **125**, 16629 (2021); Fuentes-Rodríguez, L. et al., *Mater. Chem. Front.* **6**, 2284 (2022)). But in summary, no, the voltage between poles cannot be measured directly through connected wires, without discharging the induced dipoles.

On the other hand, reference electrodes show the voltage of a nearby working electrode, via a connection to a potentiostat, which turns out to be a measurement of the solution near the material. More details are given in Fuentes-Rodríguez, L. et al., *J. Electrochem. Soc.*, **169**, 016508 (2022). This aspect complicates the study of bipolar electrodes, but other ways do exist: in bipolar electrochemistry, the reactions occurring may be the key factor suggesting which potentials are reached. In parallel, modelling the initial conditions has proven to reach very representative values, and for that reason COMSOL calculations were made in this work (see Figure S5).

A sentence has been included in the manuscript to clarify that point (page 4, line 101). A direct measurement of the induced dipole is not possible through direct contact, since the dipole discharges, but it has been evaluated previously using an indirect approximation²⁴, which agrees with the simulation shown here through finite element methods (see Figure S5).

The author's proposed mechanism requires CoN to be metallic, a point which should be made explicit in the text.

Actually, the correct requirement for an electrode, connected or unwired like the one here, is to be “conducting”. A semiconductor like carbon, but also LiCoO₂ in Li batteries, which are semiconducting, can be used as electrodes as well. Thin layers on conducting materials also work, since the macroscopic conductivity is sufficient, and so thicknesses do not involve passivation (like natural Cu₂O on Cu).

A sentence has been included in the manuscript to clarify this issue (page 2, line 54 and line 59)

Already written: that the surface of electrically conducting objects immersed in liquid electrolytes can become polarized under

In recent years, BPE has gained renewed attention for electrosynthesis of novel materials^{15,16}, and for its potential in applications such as sensing, screening and biological actuation^{15,17,18}, involving processes not only at the surface but also within the material if intercalation/deintercalation and ionic motion does exist.

The authors should revise Figure S5, the Comsol model, as it is it took a long time to interpret. As it is in the supplement, the authors could include three panels: The electric field through the sample, through the solution, and the difference. As it is, it looks like there is a changing voltage in the metallic sample (and hence an electric field), which is not possible.

We understand the referee refers to former Figure S4 (Figure S5 in the revised version of the manuscript). Thanks for pointing this out. Certainly, we can draw the electrolyte profile, the one on the sample, and the difference. We have included here a new panel in Figure 1 (Figure 1b) in the main manuscript, that clarifies the scheme. Simultaneously, Figure S5b shows the actual difference between the potential distribution and that of the electrolyte. We certainly hope it is clearer now. The inner conducting material is not charged in a simple scheme, although we would like to remind here that if intercalation is possible the actual area of the electrode may affect the bulk, as described in the Introduction. In the case of mixed conducting materials that undergo intercalation, those “surfaces” are more complex and the material itself acts as an exposed interface, thanks to the chemical diffusion. Again, semiconducting character is sufficient for the process to work and all references included in the revised version of the manuscript prove that. The following sentences have been added to the revised version of the manuscript (page 4, line 95):

“Fig. 1b depicts the potential profiles for electrolyte voltage drop, the distortion created for the immersed conducting material, and the resulting induced potentials at the poles. The interfacial potential difference between the CoN and the electrolyte solution, which is the driving force of electrochemical reactions, varies in this configuration along the lateral length of the actuated film^{22,23}.”

“b, Schematics of the potential profile for the electrolyte and the distortion occurring when a conducting material is immersed in the electrolyte. The deviation from the original electrolyte profile corresponds to the induced poles, opposing the external field, at the extremes of the material.” (caption of Fig. 1b).

Here is a summary of the comments that I would ask to be addressed:

(1) The authors could repeat the experiment moving the sample closer and further from the anode. If there is a uniform electric field in the solution then the voltage drop will be the same independent of the distance to the electrode. On the other hand, any chemical reaction with iodine will be strong near the anode and weak close to the cathode. This will allow the two mechanisms to be separated.

We certainly agree on this point. We had done this experiment but failed to include the corresponding explanations in the final version of the manuscript. The induced voltage depends on the distance between external electrodes, as predicted by electromagnetic equations in parallel electrode settings. If their separation distance is increased, while keeping the sample centered, the field will necessarily become lower. That is, keeping the same field for larger distances requires the application of much larger potentials and, as a result, more secondary reactions will happen. Note that, in other bipolar electrochemistry experiments, a larger sample with induced anode and cathode situated further away from each other also yielded the expected results (Fuentes-Rodríguez, L. et al., *J. Electrochem. Soc.*, **169**, 016508 (2022)).

As mentioned above, the optimal conditions were found after a screening step, which are those shown in the manuscript. The effects depend directly on the electric field used, and the manuscript summarizes the optimal conditions.

Since no direct reactions were observed, neither a M_S change was seen in samples with no electric field treatment, it was clear that the effect was induced by the electric field.

Also, the induced anode in the opposite pole of CoN reduction, will form I_2 , as mentioned above, so, pulling apart the Pt driving electrodes does not prevent the formation of I_2 in the media. In any case, XPS and M_S show that there is not contamination from iodine neither direct reaction with I_2 .

We have clarified the sentence related to this issue on page 15, line 284 (optimization process). “External driving voltages were applied through a power supply (Agilent B2902A) across the connected Pt electrodes immersed in the bipolar electrochemical cells. The voltage values presented throughout the manuscript always refer to external driving voltage values. After a careful optimization of distances between electrodes and sample, the Pt electrodes were distanced 20 mm from each other for the vertical BPE alignment and 25 mm for the horizontal device configuration. Thus, the applied field corresponds to $\Delta V/20$ and $\Delta V/25$ Volt/mm in vertical and horizontal configurations, respectively.”

(2) The authors could repeat the experiment with the electrodes separated from the ionic liquid, such as outside of a beaker, which will disallow any KI breakdown, but should still allow the electric field to pass through.

Again, we are not sure what the referee means and we apologize for that in advance. By definition, the electrodes cannot be outside the liquid, otherwise they cannot be considered electrodes in the purely electrochemical sense. If outside the liquid, the electrodes would be part of a capacitor that requires a different setting, much larger voltages, etc.

If the referee means using a salt bridge to protect the main electrolyte from reactions occurring at the Pt electrodes, still, KI oxidation towards I_2 occurs also at the induced anode in the sample, in a small amount. In other words, the reaction does exist and, in fact, allows the other pole desired reaction. In electrochemistry we must always have both poles transferring charge. The trick is choosing which one is

appropriate for our system and dealing with the inconveniences of it. Furthermore, an insulator like the salt bridge may screen the electric field and prevent full polarization.

The main point, however, is not to prevent that reaction since it has been shown not to modify our material, but be aware of it and use it for our purpose (it facilitates the opposite pole reaction, namely, CoN reduction).

(3) Use a reference electrode and quantify the electric field in the solution, then compare that electric field to the contact voltammetry curve.

The electric field across the cell, when only the electrolyte is present, is easily quantified by the imposed voltage, as COMSOL calculation predicts. But the existence of a conducting material generates a distortion, that can also be calculated and shown in Figure S5 in terms of voltage and charge, in a bi-dimensional manner.

A reference electrode will only work for measuring a wired working electrode, as mentioned above, direct connection of the CoN material will discharge the dipole. Please see: Fuentes-Rodríguez, L. et al., *J. Electrochem. Soc.*, **169**, 016508 (2022). So, it is not a possibility in bipolar electrochemistry since it discharges the whole process. An alternative rough approximation may be done by connecting only one wire to the bipolar electrode and approach another wire near the sample as shown in Fuentes-Rodríguez, L. et al., *J. Electrochem. Soc.*, **169**, 016508 (2022), but the diffusion and charging of the double layer involves a significant amount of noise, and we found electrostatic calculations to be pretty accurate and close. For that reason, the COMSOL profile has been included in Figure S5.

4) Look for Iodine in the EELS results (preferably when fresh from the solution, decomposition to I₂ will result in gaseous emissions and may not be present in the sample if it sits around)

XPS general survey had been run and it is now included (see Figure S8a). No evidence of Iodine 3d signals (615 and 630 eV) is present.

(5) Look for changes in the sample by submerging it in a solution of KI+I₂+PC+H₂O without an electric field

Blank samples have been measured and are shown now, including *M_s* studies. No magnetic effect is observed either, please see newly added Figure S4 with a signal equal to that of the as prepared sample.

A sentence has been included in the manuscript clarifying that (page 3, line 92, and supplementary information Fig S8a). Note that neither visible chemical reactions nor magnetic properties changes are observed upon immersion of the CoN in the electrolyte with the absence of external applied voltages (Section 4, Supplementary Information).

(6) In the horizontal configuration, the authors should measure opposite ends of the sample as well as the middle.

As described above, a new experiment has been carried out in which the horizontal treated sample has been split in three parts, as the referee suggests, corresponding to the negative poles, the center and the positive

pole. Indeed, a different M_S is found for each part, as represented now in Figure 1c. A gradient in M_S does exist, with the negative pole showing a much larger M_S value (Figure S6). It is necessary to note here that because the gradient evolution is dynamic, and the measurements are sequential, the absolute magnitudes have a larger error. But still the gradient remains after the first two hours, as expected from previous observations. Such experiment confirms our hypothesis of the redox gradient existing in the horizontal configuration sample.

The same dynamic evolution is evidence of the gradient, as described in other systems (see: Fuentes-Rodríguez, L et al., *J. Phys. Chem. C* **125**, 16629 (2021)), and explain why long-range order techniques are not sensitive (see: Fuentes-Rodríguez, L. et al., *Mater. Chem. Front.* **6**, 2284 (2022)).

Minor Comments:

I was generally unhappy about this statement: "...subsequent structural characterization of the treated samples was hindered due to the volatile nature of the induced effects in the horizontal BPE treatments" Why was it so much more volatile compared to the vertical configuration?

As it was explained in the original version of the manuscript, there is a gradient in charge along the CoN layer, in the direction of the electric field. It is obvious that the negatively charged becomes a cathode, while the positive one becomes an anode, and such charge gradient induces a chemical gradient (electrochemical gradient). The resulting material has a charge/oxidation state gradient, and once removed from the electrolyte the existence of a gradient is a driving force for homogenization, as previously observed in other systems [Fuentes-Rodríguez, L et al., *J. Phys. Chem. C* **125**, 16629 (2021)]. In the horizontal arrangement, a clear gradient exists and can be shown thanks to colored indicators or in certain systems. Here, the Co color is evident at the negative part but not the positive or the center, despite being conducting, since the interface with the electrolyte is subject to different potentials as shown in [Fuentes-Rodríguez, L. et al., *J. Electrochem. Soc.*, **169**, 016508 (2022)] and COMSOL calculation.

The vertical configuration, however, has all the surface exposed at the same distance to the electrodes and thus, it is subject to the same potential. Only across the width a gradient, very small (note that the sample is thin), could appear.

The new VSM measurements, done on three different portions of the sample, following the referee's suggestion, shows it also clearly (See Figure S6), confirming the existence of the gradient, as observed in other systems using other techniques.

We have clarified the existence of the gradient further in page 6, line 125. "...samples was hindered due to the time evolution of the gradient observed in M_S and therefore the derived chemistry, and the impossibility to perform all experiments in the same time period, that is, due to the volatile nature..."

The following sentence makes an important point that should be emphasized earlier, that the horizontal configuration only works near the ends of the sample:

"The volatility of the effect in the horizontal configuration was the occurrence of a redox/ionic lateral gradient within the zone where potentials are sufficiently high. Such gradient is likely to be present also for the vertical configuration (perpendicular to the sample), but with much smaller magnitude since the film is very thin and now the entire outer surface of the film (and not only the edges) contributes to the redox process."

We agree with the observation of the referee, and we have explained the effect earlier in the text (page 6, line 125). "...samples was hindered due to the time evolution of the gradient observed in M_S and therefore the derived chemistry, and the impossibility to perform all experiments in the same time period, that is, due to the volatile nature..."

"X-ray photoemission spectroscopy (XPS) study (Section 5, Supplementary Information) shows a decrease of the nitrogen content for the treated sample, thus corroborating the voltage-induced nitrogen ion release to the electrolyte" This just shows a decrease in valence, which is consistent with nitrogen extraction, but also iodine reaction.

The atomic ratio N/Co shows the decrease of N content in the sample. It has been included now in Table S2. We agree that other reactions could be potentially involved in the reduction, but the evidence shows they are not. Iodine is not observed by XPS, so, it cannot be a reaction with iodine. The manuscript contains now a clarification about the absence of iodine, both in blank immersed samples and voltage-treated samples, including Figure S8a and text in the main text (see above) and supplementary material. The sentence has been rewritten in the revised version of the manuscript (page 9, line 200):

"Based on previous literature on cobalt nitride with variable nitrogen concentration³¹, the X-ray photoemission spectroscopy (XPS) study (Section 8, Supplementary Information) shows a relative change on the N/Co ratio and on the N components for the treated sample, thus corroborating the voltage-induced nitrogen ion release to the electrolyte."

The authors should report their leakage current. Also, does the leakage current change with the sample in versus not? I ask because if there is sufficient electrical conductivity, then this becomes not so much wireless, as much as current transport through the solution.

We thank the referee for pointing this out. Certainly, any electrochemical cell involves a conducting electrolyte, and there the conduction is ionic. For that reason, we are defining the process as "wireless", instead of contactless. The effects on the material are through the electrolyte and not through direct wire contact as previous papers have reported.

The current registered/observed is the current through the connected Pt electrodes, and not through the bipolar electrode, which cannot be connected. When electrode reactions occur, the current is an exponential decrease, as expected. Part of the conduction is governed by Ohms law and the electrochemistry causes a deviation from it, in agreement with fundamental electrochemical equations. We have included the current versus time graph in Figure S9, and discussed it there (page 15, line 299):

"*I vs. t* curves for the connected Pt driving electrodes during the bipolar experiments are shown in the Supplementary Information, Section 9."

In order to clarify that beyond for this case, we have demonstrated previously [Fuentes-Rodríguez, L. et al., *J. Electrochem. Soc.*, **169**, 016508 (2022)] that the presence of conducting materials immersed in electrolytes lowers the impedance of the system. Conduction does occur thanks to the existence of the electrolyte, and that is what the system is based on. Polarization due to the existence of the conducting part is what modifies the profile of voltage as described in Figure 1b and Figure S5 in the main text and Supplementary Information, respectively.

In summary, for the same potential, current is larger if conducting materials are used (given that change in Z), which involves substantial applications that are being studied right now. And the electrochemical

processes may be considered a “leakage” of current, since there is a transfer between electrode and solution. Here, in our case, only the current passing through the Pt driving electrodes can be recorded.

REVIEWER COMMENTS

Reviewer #1 (Remarks to the Author):

The authors carefully replied to all my concerns and comments. The response are adequate and wherever reasonable and possible data has been added. As a consequence, in my opinion the manuscript should be published at this stage as my criticism have been addressed.

Reviewer #2 (Remarks to the Author):

In their response, the authors compare the significance of their work to what they refer to as 'wireless techniques': ion irradiation and light-induced switching. Concerning light ion irradiation, the seminal publication by Chappert et al. clearly demonstrates its direct potential for applications in magnetic patterning in nanoscale devices. The papers that the authors cite concerning light-control of magnetism present a completely new mechanism for controlling magnetic states through light-wave-induced coherent transfer of spin, which goes far beyond known light-spin interaction mechanisms like all-optical switching. Unfortunately, the study presented here neither provides a demonstration of a new potential application made possible by their method, nor introduces a novel physical/chemical mechanism for controlling magnetic states, as is the case with the aforementioned publications.

I would like to clarify that I do not intend to imply that this work is unworthy of publication. The research conducted is of good quality, thorough, and interesting to the scientific community. However, I believe it does not introduce anything substantially new beyond a new method to achieve effects that are already known, namely, the volatile/non-volatile electric field control of magnetic order through ionics.

Notably, two recent publications in magneto-ionics in Nature Communications present: (I) the magneto-ionic-driven sign change of the Dzyaloshinskii–Moriya interaction in solid-state devices and its use to reverse the current-driven trajectory of skyrmion motion (Fillion et al., Nature Communications 13, 5257 (2022)), and (II) the solid-state magneto-ionic control of the Spin Seebeck effect, enabling a tunable one-order-of-magnitude enhancement of the thermoelectric signal (J.-M. Kim, Nature Communications 14, 3365 (2023)). In my opinion, it is important to acknowledge the recent advancements in this field, and I consider other works on magneto-ionics currently being published in the same journal to have a greater impact.

In their response, the authors repeatedly emphasize the potential of their work for various applications. As I mentioned in my previous report, a significantly greater impact and originality could have been achieved by using this method to present a demonstration of something that could not be done using physical electrical connections to the sample. However, the authors have not tried to present even a simple proof of concept for any of these claims, weakening their argument. Unfortunately, I still believe that this work lacks significant novelty, and therefore, I cannot recommend it for publication in Nature Communications.

Reviewer #3 (Remarks to the Author):

The comments provided in the last round have been addressed in this revision. I am ok to move forward on this version for the publication.

Reviewer #4 (Remarks to the Author):

The authors did not effectively respond to my comments. In light of two citations that the authors repeatedly reference, I continue to be skeptical of the scientific validity of these results. The authors repeatedly cite articles by Fuentes-Rodriguez, and these articles directly contradict several of the claims by the author and undermine their justification for not performing subsequent

experiments. Here are some examples:

The potential cannot be measured, contradicted by J. Electrochem. Soc., 169, 016508 (2022) Figure 4. This is actually interesting, because Fuentes-Rodriguez states "direct voltage measurement of the dipole is not possible because the connections would undergo the same bipolar phenomena" (NOT a discharge effect), but then perform a very analogous measurement (Fig. 4 in this text). In the geometry of Figure 4, I think that touching either end of the sample (outside of the electrolyte) would provide a valid voltage. As I asked before, please perform this experiment (one or the other).

The system needs to be optimized, contradicted by J. Phys. Chem. C 125, 16629 (2021), Figure 9 and associated discussion: "When using one, two, or three stripes, the change in energy corresponding to the dipole is equal among them, and stripes become different in the stripes closer to the driving electrodes and only at high voltages when O₂ and H₂ are formed. Therefore, the position within the parallel electrode field is not affected in terms of bipolar electrochemistry effects." This was my exact concern, products being generated at the electrodes! Also J. Electrochem. Soc., 169, 016508 (2022) Figure 3, which shows the effect is independent of position. In both of these papers they effectively perform the experiment I asked for. It seems a standard and acceptable request. As I asked before, please perform this experiment.

~~~~~  
~~~

Also, the authors did not (as far as I can tell) perform the proposed emersion experiment. From the response, the authors submerged the sample in PC+KI, this is not what I asked for, rather PC+KI+I₂. KI+I₂ is a well known metal etchant, KI is not. Having not performed the proposed experiment, I this comment was not addressed. As I asked before, please perform this experiment.

My discussion of enthalpy should have motivated the authors to estimate the entropy and perform a Gibbs free energy calculation, to see if this happens spontaneously at room temperature. It does, it is thermodynamic fact, it was not a suggestion. The next question the authors would ask is 'why don't we see it in EDX', and a little bit of work would reveal that Co₂I is highly soluble in water and other polar solvents (including PC), comparable to table salt. So, it is no surprise that they don't see anything in EDX/XPS, is it no longer part of the film. This is something the authors should have done.

I found the discussion on the optimization very discouraging, as they are fairly flawed. "...we have given the optimized field distance setting. Before that, several distances and voltages were screened (of course, the larger the distance, the smaller the external field for the same voltage)." Also "The induced voltage depends on the distance between external electrodes, as predicted by electromagnetic equations in parallel electrode settings." Again, see Electrochem. Soc., 169, 016508 (2022) Figure 3 and J. Phys. Chem. C 125, 16629 (2021), Figure 9, there should be no electrochemical dependence on the position between the electrodes. Furthermore, from a fundamental perspective you can understand this two ways: the 'physical' way, the electric field is uniform between the electrodes, thus the polarization of the film will be the same, independent of the sample position. The analytical way to think of this is that the induced voltage is the integral of the electric field over the length of the sample; since the electric field is uniform between the electrodes, and the length of the sample is position independent, the voltage across the sample is position independent. The authors use the term 'distance' from the electrode, inherently ignoring that the electric field starts at one electrode and ends at the other, this isn't some decay function. So based on this fundamental understanding and discussion in Electrochem. Soc., 169, 016508 (2022), if the authors need to optimize a position, this is not an electrochemical effect.

The authors seem confused about the operation of their electrochemical cell. Generally, electrodes are placed in an electrolyte and a voltage applied, generating an electric field between the electrodes, through the interstitial dielectric (or IL in this case). A second effect occurs for the case of higher voltages, namely that the electrode reduces/oxidizes the material (as is the case in e.g. batteries or electrolysis cells), through the exchange of electrons with the dielectric/IL. Here, the authors are adamant that this is an electric field only effect – the electric field polarizes the sample, which generates an EDL, which then extracts ions from the ionic film – and not electrochemical. By moving the electrodes outside of the IL, the electric field still exists, penetrates the glass (obviously), should still polarized the sample, generate an EDL, and magneto-ionic migration would occur. The difference is that there is now no electron exchange at the electrode, e.g. no reduction or oxidation. The authors comment about the electric field, but the electric field is simply V/d , which is the same for the same electrode spacing. Put another way, if this is truly

wireless in the way the authors describe, then electron exchange is not necessary, so remove the source of electrons and see if it still works. So, again, please perform this experiment.

I like the author's new Supplemental Figure S6, and appreciate the discussion up to the last sentence: "Such experiment confirms our hypothesis of the redox gradient existing in the horizontal configuration sample and discards any possibility of a direct chemical reaction.". If this is iodine-based reduction, it would the results would show a gradient based off the diffusion of iodine from the cathode, so I don't agree with this interpretation. I also disagree with the author's claim of a discharge here, they really measured 3 points and have no resolution if it is time or spatial. If we look at the author's previous measurements (Nature Communications 11, 5871 (2020)), for a discharge, the authors should have seen the magnetization continuously decreasing during the measurement, which they do not see here. If the authors want to make the claim that there is a discharge effect, they should (1) measure the magnetization versus time, or (2) re-measure a new sample in a different order (e.g. middle, end, end) and show that it has the same decreasing trend. Otherwise, the claims of discharge should be removed.

In Summary, Last time I asked for PI+KI+I2 submerged experiment, please do this. Last time I asked for position dependence (as seems standard in the other published works), please do this. Last time I asked for a potential measurement, the authors citations show how to do this, please do it. Last time I used gentle language on the thermodynamic calculations, but this is fundamental fact and cannot be disputed, please do not disregard it. Last time I asked for an experiment with an electric field only (by insulating the electrodes so there can be no charge transfer), the authors did not do this, please do it. Last time I asked for measurements at 3 places, this was done, but now is accompanied by new, unsupported claims of discharge, please perform time dependent measurements on a single sample to confirm, or remove it.

RESPONSE TO REVIEWERS' COMMENTS

Reviewer #4

(Please note that the referee's comments are shown in black font and our responses in blue)

The authors did not effectively respond to my comments. In light of two citations that the authors repeatedly reference, I continue to be skeptical of the scientific validity of these results. The authors repeatedly cite articles by Fuentes-Rodriguez, and these articles directly contradict several of the claims by the author and undermine their justification for not performing subsequent experiments.

First, we sincerely appreciate the comments that the referee made in both review rounds, since they helped us to identify possible misinterpretations. After the first review round, we took into consideration and tried to perform all suggested experiments to prove or disprove each concern. However, some of the requested experiments are not technically feasible. For example, 1) voltage is always measured between two points, as fundamental physics dictates and, hence, it cannot be determined using only one pole; 2) electrodes in an electrochemical cell cannot be removed from the electrolyte without jeopardizing the whole experiment.

Having this in mind, in the following, we explain why there is no contradiction with our previously published studies.

Our previous *J. Electrochem Soc.* paper explains exactly that no direct voltage can be measured in the induced dipole and suggests an alternative approximation by leaving a lead close to but not touching the material. This introduces large errors in the values of the estimated voltage because that lead unavoidably undergoes polarization.

Additionally, we would like to remind that "potentials" in electrochemical terms are directly related to ΔG , that is, $\Delta G = -nFE$. Therefore, the observed potential is already a combination of enthalpy and entropy factors. Any meaningful reasoning involving entropy would require considering structural features for the solid material, which are not precise at the nanoscale, and incorporating *ab initio* calculations at the nanometer scale that are much beyond the scope of the manuscript.

Finally, the applied field certainly depends on the distance between the driving electrodes. Once this is fixed, the immersed material (acting as bipolar electrode) may be placed in different positions between the driving electrodes, in the region of high intensity field, obtaining the same induced dipole. Leaving the physics aside, as you get close to the driving electrodes you also have oxidation products that may reduce at the induced cathode of the bipolar electrode, and reduced products that may reoxidize at the induced anode. Both cases would lower the charge efficiency of the process (e.g., CoN reduction will be less efficient if I_2 also reduces at the same time in the induced cathode).

Here are some examples:

The potential cannot be measured, contradicted by *J. Electrochem. Soc.*, 169, 016508 (2022) Figure 4. This is actually interesting, because Fuentes-Rodriguez states "direct voltage measurement of the

dipole is not possible because the connections would undergo the same bipolar phenomena" (NOT a discharge effect), but then perform a very analogous measurement (Fig. 4 in this text).

We appreciate the concerns that the reviewer points out. However, we would like to note that we are the authors of this paper. In such reference, the fact that the direct induced potential cannot be measured was clearly stated. Then, an estimation was made by touching only one end of the sample and approaching the other contact. But this was only a rough approximation. As it was explained in the paper, the wet contacts also undergo a bipolar effect, introducing a significant error to the approximation. Thus, two aspects must be considered: 1) it is not possible to contact both poles directly because this causes a discharge of the induced dipole, and 2) by approximating with one contact inserted in the electrolyte, the error is large because the lead also undergoes polarization. In any case, an induced voltage does exist, even if not quantified experimentally, and the best indication is its existence are the redox reactions at the poles (as this paper also shows in the pictures), since redox reactions are expected to occur at the induced cathode and anode.

Thus, overall, we do not contradict the reported reference (our previous work).

In the geometry of Figure 4, I think that touching either end of the sample (outside of the electrolyte) would provide a valid voltage. As I asked before, please perform this experiment (one or the other).

We can understand the puzzle where the reviewer tries to contribute. However, we must recall that voltage is always a difference between two poles, by intrinsic definition of voltage. Touching both ends discharges the dipole, while touching one end requires the ionic transmission, as done in our previous *J. Electrochem. Soc.* paper. The latter constitutes a measurement with large associated error. It will never be a "valid voltage" as the reviewer claims. In our opinion, measuring pseudo-voltage values using this approach does not bring anything to the current manuscript. The best indicator of the built-in voltage is the occurrence of redox reactions.

The system needs to be optimized, contradicted by *J. Phys. Chem. C* 125, 16629 (2021), Figure 9 and associated discussion: "When using one, two, or three stripes, the change in energy corresponding to the dipole is equal among them, and stripes become different in the stripes closer to the driving electrodes and only at high voltages when O₂ and H₂ are formed. Therefore, the position within the parallel electrode field is not affected in terms of bipolar electrochemistry effects." This was my exact concern, products being generated at the electrodes!

Indeed, we agree with the reviewer on the existence of other reactions, as we reported in some of our previous papers, and also in this manuscript. Electrochemical reactions occur at the driving electrodes and at the induced poles in the unwired electrode. Our new manuscript does not claim otherwise. In fact, it tries to pinpoint which reactions occur in this new system, and which ones are responsible for the observed magnetic changes.

Again, this does not contradict prior observations reported on the *J. Phys. Chem. C* paper. In that case the electrolyte was aqueous, and at certain potentials, H₂ and O₂ were formed at the driving electrodes. In addition, iridium oxide in the bipolar electrode underwent redox changes too, and that was measured very clearly through XAS experiments. Different ranges of driving voltages existed, where the extent of the reactions varied in the Pt driving electrodes. But this did not prevent the induced reduction of IrO_x from occurring, which allowed the concomitant neural cell growth.

In our current manuscript, we have explained that I_2 is formed both at the driving anode and at the induced anode pole. The existence of such I_2 formation is also a driving force for the reaction occurring at the induced cathode (CoN reduction).

We have also explained that no direct reaction between Co and I^- is observed (neither in previous papers with cobalt phases in iodide media).

It is true, though, that we failed to explain further that I_2 approaching the induced cathode will immediately reduce back to I^- , as the cyclic voltammetry suggests. But if a reaction between Co and I^- would happen, Co would be depleted from the sample, forming blue green CoI_2 , and this was not observed (please note that CoI_2 is a color indicator for water and both, blue and pink may be observed depending on the amount of water). Furthermore, because potentials (E) are related to ΔG , such spontaneous reaction would modify the potential measured in solution, and that was never observed (starting potentials in cyclic voltammetry experiments are very stable). To clarify this issue, we have included all these points in the revised version of the manuscript (page 6 marked in green):

“Alternative secondary reactions, like Co metal (formed by reduction) oxidized back by I_2 yielding soluble green CoI_2 , do not seem to take place, since no Co depletion occurs (*i.e.*, no green colour is observed and M_s increases). No iodine signal is observed either in the solid material, so no additional phases containing iodine form, in agreement with previous works using this electrolyte²⁸.”

We would like to emphasize that we screened all possible reactions that could be responsible for the observed increase in M_s . None of the possibilities suggested above can explain the remarkable enhancement of M_s in the solid material. Previous studies under direct contact conditions, also using halide salts, confirm our interpretation (see e.g., S. Martins et al. *Nanoscale Horizons* 2023, 8, 118) using various electrolytes). Moreover, the observed changes in the magnetic response of the bipolar electrode CoN indicate unambiguously the reduction of CoN, since only such reduction can explain the M_s enhancement. This is in agreement with the XPS and EDX data included in the last review and the absence of solution color changes.

Also J. Electrochem. Soc., 169, 016508 (2022) Figure 3, which shows the effect is independent of position. In both of these papers they effectively perform the experiment I asked for. It seems a standard and acceptable request. As I asked before, please perform this experiment.

We agree with the reviewer that the potential does not change with position if the distance between the main driving electrodes is not varied. But the proximity to the redox reactions occurring at the driving electrodes may lower the efficiency of CoN reduction (I_2 re-reducing at the induced cathode and using part of the charge transfer). Bearing this in mind, the optimal positioning was searched in our study (and then, fixed) to maximize M_s .

We recall that in our paper published in J. Electrochem. Soc., we performed the experiment the reviewer mentions because we dealt with aqueous IrO_x experiments. Indeed, we could increase the distance between the Pt driving electrodes to accommodate additional positions without such interference from secondary reactions. However, with propylene carbonate we need larger voltages that in turn magnify the driving electrode reactions, enhancing I_2 and gas contents to non-manageable points, where the Ti adhesion layer tends to oxidize and the CoN coating detaches along with Ti and

Au. Thus, such high degree of optimization that we could do in our previous J. Electrochem. Soc. or J. Phys Chem C papers cannot be achieved here without cell rupture, i.e., there is a limitation in charge delivery due to low PC electrolyte conductivity. In other words, the nature of the solvent imposes restrictions in the voltage that can be applied. In fact, as noted in the main text of the manuscript, there is the detachment the CoN coating above 20 V driving voltage.

In summary, one thing is the distance between driving electrodes, and the other the distance between bipolar unwired sample and the driving electrode. A change in the latter does not modify the induced potential, but a change in distance between the driving electrodes does change the required applied voltage and a larger voltage involves solvent decomposition and detachment of the coating.

Also, the authors did not (as far as I can tell) perform the proposed emersion experiment. From the response, the authors submerged the sample in PC+KI, this is not what I asked for, rather PC+KI+I₂. KI+I₂ is a well known metal etchant, KI is not. Having not performed the proposed experiment, I this comment was not addressed. As I asked before, please perform this experiment.

We certainly appreciate the discussion with the reviewer on this counterintuitive subject, and we are aware it is often not well understood.

Emersion is fundamentally against electrochemistry. If the electrodes are not immersed in the electrolyte, there is no electrochemistry due to the absence of charge transfer at the electrode-electrolyte interface.

We modestly disagree with the comment that we have not done the requested experiment. In the first review round, the reviewer suggested both immersion in KI and KI and I₂. But the experiments in PC+KI at low voltages already proceed with I₂ production at the driving electrode, and this does not cause any change in magnetism in the solid sample CoN. Neither a color change was observed (CoI₂ would give a clear change of color) near the sample. We have clarified this aspect in the text related to possible reactions involved (page 6 marked in green), as indicated above.

The plot of M_s vs. voltage is an indirect proof that no reaction happens between I₂ and CoN or Co at low voltages, for which we already have I₂ at the driving electrode. Nor soluble green CoI₂ is formed and dissolved. Finally, no iodine signal was detected by XPS or EDX experiments in the solid, as seen from Fig. S8 of the supporting information and our last round response letter.

My discussion of enthalpy should have motivated the authors to estimate the entropy and perform a Gibbs free energy calculation, to see if this happens spontaneously at room temperature. It does, it is thermodynamic fact, it was not a suggestion.

We certainly did not want to enter into entropy discussions, although tempted after reading the reviewer's comment. There is a fundamental reason for it: E values in electrochemistry are directly related to ΔG (i.e., $\Delta G = -nFE$), where n is the number of electrons exchanged, F the Faraday constant and E the electrochemical potential. So, the potentials required or obtained for a process when direct contact is made or when E is induced, determine the value of ΔG . In the case of measuring rest potentials, a spontaneous direct reaction would change such open circuit voltage (as in titrating of MnO₄²⁻ with Fe²⁺ for example). This does not occur here, and the cyclic voltammetry shows a stable initial potential (I₂ is already present in the solution) and also a good reproducibility for the overall

range of potentials (I^- and I_2 and reduced Co). Therefore, no spontaneous reactions occur between Co and I_2 or between CoN and I^- .

In terms of the reviewer discussion, enthalpy data concerns calculations that are based on specific structures for Co, CoI_2 , etc. Those conditions may be far from the ones found in propylene carbonate (including solvation and crystallization, and charge transfer), and also in solids at the nm scale, with a room temperature treatment that prevents good crystallization. We consider a ΔH and ΔS discussion is not appropriate for the manuscript, even if it may have some interest, but the E values already relate to empirical facts (no enhancement of M_s would be observed if cobalt would be dissolved). A sentence has been included in the revised version of the manuscript to emphasize on the stability of open circuit potentials, implying the absence of spontaneous direct chemical reactions of any type, in the solid or in solution (page 6-7, marked in green):

“Open circuit potentials and initial voltammetry potentials are very stable and constant, evidencing no spontaneous direct reactions in the setting.”

The next question the authors would ask is ‘why don’t we see it in EDX’, and a little bit of work would reveal that Co_2I is highly soluble in water and other polar solvents (including PC), comparable to table salt. So, it is no surprise that they don’t see anything in EDX/XPS, is it no longer part of the film. This is something the authors should have done.

We are thankful to the reviewer for this comment. As mentioned before, CoI_2 dissolved in the solution would have given a change of color near the sample (CoI_2 is blue-green in the absence of water and pink if water is present and it is typically used as an indicator), which was not observed. Furthermore, I_2 formed at the driving anode, would re-reduce back at the induced CoN cathode, even before reacting with Co. Even more, if I_2 would leach metallic cobalt, no enhancement of M_s would have been observed. All Co would eventually disappear from the sample, and no ferromagnetic signal would remain.

Again, we appreciate the suggestions made by the reviewer on several possibilities, since it is desirable to disclose the mechanism in which iodine or iodide may take part. But none of those possibilities explain the observed dramatic increase in M_s , neither in direct electrochemistry nor here, in induced bipolar electrochemistry experiments. We certainly considered the reviewer’s suggestion as something possible, but no evidence of spontaneous formation of CoI_2 was found (from either the solution or the solid), neither from the characterization techniques used nor from the observed stability of potential or the color change.

I found the discussion on the optimization very discouraging, as they are fairly flawed. “...we have given the optimized field distance setting. Before that, several distances and voltages were screened (of course, the larger the distance, the smaller the external field for the same voltage).”

Thanks again for raising this remark. Indeed, using larger distances between driving electrodes gives more room for achieving variable values of induced voltages and an easier charge transfer. However, here, with propylene carbonate, with smaller conductivity than water, the window in which the solvent does not decompose is also smaller, therefore we cannot tune the distance so much. Within the available empirical voltage range, we have optimized the maximum M_s value achieved, with no

interference from I_2 at the driving anode that could lower the efficiency of the reduction process in the induced CoN cathode (I_2 would reduce before CoN, thus decreasing the efficiency).

Also "The induced voltage depends on the distance between external electrodes, as predicted by electromagnetic equations in parallel electrode settings." Again, see *Electrochem. Soc.*, 169, 016508 (2022) Figure 3 and *J. Phys. Chem. C* 125, 16629 (2021), Figure 9, there should be no electrochemical dependence on the position between the electrodes.

We are thankful to the reviewer for this remark, but please note that we have TWO distances in our experiments: 1) the distance between the driving connected Pt electrodes which define the electric field. 2) the position of the sample with respect to the electrodes (see Figure 1 of the manuscript). Once you fix a particular distance between the Pt electrodes, moving the unwired electrode between the driving electrodes does not modify its polarization (that is, the potential among borders in the unwired material). The induced dipole is the same if one does not get out of the main field lines. However, one may approach zones of larger concentrations of formed I_2 if one gets too close to the anode. Then, as explained above, the produced I_2 at the driving anode may reduce at the induced cathode in CoN, lowering the efficiency of the process.

No contradiction exists with our previously published results in the references cited by the reviewer. We did observe the same effects with distances, only we had a different window due to dissimilar conductivities of the electrolyte.

Furthermore, from a fundamental perspective you can understand this two ways: the 'physical' way, the electric field is uniform between the electrodes, thus the polarization of the film will be the same, independent of the sample position. The analytical way to think of this is that the induced voltage is the integral of the electric field over the length of the sample; since the electric field is uniform between the electrodes, and the length of the sample is position independent, the voltage across the sample is position independent.

That is certainly true if the distortion induced by the conducting material is not taken into account. Within the metal, electric field does not exist, but at the borders where the metal ends the sample gets polarized and forms a dipole of opposite sign to the imposed field (see Figure 1 and S4). So, the voltage profile changes when a conducting sample is present, as Figure S4 shows, and the voltage profile across the sample also shows that the integral of the field is not described by the simple imposed field. It is for this reason that induced unwired electrochemistry appears.

The authors use the term 'distance' from the electrode, inherently ignoring that the electric field starts at one electrode and ends at the other, this isn't some decay function. So based on this fundamental understanding and discussion in *Electrochem. Soc.*, 169, 016508 (2022), if the authors need to optimize a position, this is not an electrochemical effect.

The electric field in the absence of the sample is indeed constant between two parallel electrodes. Then, a distortion is caused by the polarization of the conducting sample (see the voltage profile in Fig 1b of the manuscript). But when we talk about distance it is because there is a voltage (not field) profile. Voltage decreases in the electrolyte from the positive to the negative electrode, and the material creates an opposing effect of different magnitude.

The position needs to be optimized because reaction products formed at the driving anode (I_2 here) may be reduced at the induced CoN cathode, thereby decreasing the efficiency of the process, as explained above. So, it is mainly an electrochemical effect.

The authors seem confused about the operation of their electrochemical cell. Generally, electrodes are placed in an electrolyte and a voltage applied, generating an electric field between the electrodes, through the interstitial dielectric (or IL in this case). A second effect occurs for the case of higher voltages, namely that the electrode reduces/oxidizes the material (as is the case in e.g. batteries or electrolysis cells), through the exchange of electrons with the dielectric/IL.

We appreciate the reviewer's concern, but we believe there is no confusion here. Driving electrodes have a double layer capacitive effect and specific electrochemical reactions depending on the potential applied. Likewise, the bipolar unwired CoN electrode will experience both capacitor and specific reactions depending on the voltage at the induced anode and cathode.

Different electrochemistry reactions occur at different potentials above the threshold of the double layer potential (please see the voltammetry curves shown in Fig. S3), and always by a charge transfer at the interface between the electronic conductor and the ionic conductor. In this case, reactions occur at the interfaces of the connected driving electrodes and of the immersed unwired bipolar electrode.

Here, the authors are adamant that this is an electric field only effect – the electric field polarizes the sample, which generates an EDL, which then extracts ions from the ionic film – and not electrochemical.

Please note that we never claimed that! We say it is a capacitive plus electrochemical effect. For this reason, I_2 is formed, and also CoN is reduced to Co. A sentence has been modified in the introduction to avoid misunderstanding (page 2):

“Under application of external voltage, the electrochemistry fundamental aspects behind the formation of electric double layer (EDL) at the interface between the solid target material and an adjacent electrolyte, as well as the redox changes occurring at the ionic/electronic conducting material, are central to enable ionic migration.”

We apologize if we were not sufficiently clear on this matter.

By moving the electrodes outside of the IL, the electric field still exists, penetrates the glass (obviously), should still polarized the sample, generate an EDL, and magneto-ionic migration would occur.

We are sorry but we fully disagree with the reviewer. By removing the electrodes, one should not expect the presence of any electrochemistry. Instead, you will have an air capacitor, but if the electrodes are not in contact with the electrolyte, the possible charge transfer does not occur at the electrochemical cell, but in air. Even more, a field of this magnitude does not penetrate glass.

The difference is that there is now no electron exchange at the electrode, e.g. no reduction or oxidation. The authors comment about the electric field, but the electric field is simply V/d , which is the same for the same electrode spacing. Put another way, if this is truly wireless in the way the authors describe, then electron exchange is not necessary, so remove the source of electrons and see if it still works. So, again, please perform this experiment.

Creating an air capacitor with Pt in air creates an interface with air, not with an electrolyte and does not involve charge transfer in the electrolyte. So, no reaction will occur at the driving electrodes, but neither at the CoN. Instead, you may have a Van der Graaf electrostatic effect. Both will be wireless effects, yes, but without liquid electrolyte (IL) there is no electrochemistry.

Please note that the actual source of electrons is the material itself. CoN is reduced because of the polarization in the macroscopic piece. The electrons come from the opposite pole (where I^- is oxidized leaving electrons behind).

I like the author's new Supplemental Figure S6, and appreciate the discussion up to the last sentence: "Such experiment confirms our hypothesis of the redox gradient existing in the horizontal configuration sample and discards any possibility of a direct chemical reaction.". If this is iodine-based reduction, it would the results would show a gradient based off the diffusion of iodine from the cathode, so I don't agree with this interpretation.

If iodine does oxidize metallic Co (reduced before by electrochemical process) and migrates, we should see iodine signal in the XPS and EDX data. However, this is not the case, as mentioned above.

The only possible explanation rejects the idea of iodine or iodide migration through the sample.

I also disagree with the author's claim of a discharge here, they really measured 3 points and have no resolution if it is time or spatial. If we look at the author's previous measurements (Nature Communications 11, 5871 (2020)), for a discharge, the authors should have seen the magnetization continuously decreasing during the measurement, which they do not see here.

In the horizontal configuration we do see a decrease of M_s with time (shown in Fig 1c of the Main Text), and a gradient similar to those previously observed for IrO_x could explain the effect. The new M_s measurements after cutting the sample into three parts (Section 6 of the Supporting Information) agree with these observations. M_s measurements are extensive and cannot be reached with the same resolution as a XAS experiment, but still the conclusions match the existence of the gradient (larger M_s at the original induced cathode).

If the authors want to make the claim that there is a discharge effect, they should (1) measure the magnetization versus time, or (2) re-measure a new sample in a different order (e.g. middle, end, end) and show that it has the same decreasing trend. Otherwise, the claims of discharge should be removed.

Many thanks for this insightful comment. The effect is better described as a gradient being homogenized. The new reversibility measurements show that a physical discharge helps recover the original state. There are, in fact, two different facts: the discharge, which shows the amount of charge that is part of the capacitor component; and the gradient, which is more related with the redox state originally induced in one pole and diffusing towards the opposite pole (an electrochemical reaction). We did not include further details on the discharge because we agree with the reviewer that this aspect requires additional studies which are out of the scope of the current manuscript.

In Summary, Last time I asked for PI+KI+I2 submerged experiment, please do this.

Please see our reply above.

Last time I asked for position dependence (as seems standard in the other published works), please do this.

Please see comments above. It has been done within the range allowed in propylene carbonate.

Last time I asked for a potential measurement, the authors citations show how to do this, please do it.

Please see our reply above. Measurement of "valid" potentials in the material is not possible by direct contact or with one single contact.

Last time I used gentle language on the thermodynamic calculations, but this is fundamental fact and cannot be disputed, please do not disregard it.

Please be sure we do not disregard this. ΔG is $-nFE$ and we do speak about potential, E . Thus, enthalpy and entropy are both included. In the future, fundamental calculations may be useful to establish the ΔH or ΔS components. However, this aspect is not straightforward to be tackled when the sample is not well crystallized, i.e., after a room-temperature formation, where usually new properties may come out.

Last time I asked for an experiment with an electric field only (by insulating the electrodes so there can be no charge transfer), the authors did not do this, please do it.

Please note that insulated electrodes are not electrodes, hence no bipolar effect would exist. Please be sure this experiment only yields an "overload" of the power source. It is certainly not possible. It is also against all charge transfer. In this respect the reviewer is right, no charge transfer will occur at the Pt driving electrodes, neither at the CoN. The experiment is equivalent to a test on a blank sample. Such blank experiment has been included now and shows no effect.

Last time I asked for measurements at 3 places, this was done, but now is accompanied by new, unsupported claims of discharge, please perform time dependent measurements on a single sample to confirm, or remove it.

Please note that the first version of the manuscript included a time evolution experiment (Figure 1c). The combination of such time evolution of magnetization with the new measurements by parts supports the idea of a gradient that evolves with time. We are dealing with a gradient evolution, not a discharge. These are different terms. What we did mention is that physical discharge helps recover the sample, in the reversibility new experiment.

REVIEWER COMMENTS

Reviewer #4 (Remarks to the Author):

As with their first response, the authors have not satisfactorily addressed my concerns. If anything, the author's acknowledgement that electrode products (in the form of I₂) affect their results in an as of yet unidentified way further reinforces my belief that the authors are simply generating a solvent at one electrode and it diffuses to the sample, a very uninspiring story. As a reviewer, we are tasked to verify the scientific accuracy of a work and its appropriateness for the proposed journal. In this case I cannot do either of these, I am not convinced the work is scientifically accurate, and it very well may be intellectually trivial. I provided the authors with 5 experiments to confront my claim and they refuse to perform any of them. I therefore have no choice but to continue with my position that this work should not be published in its current form.

I have provided expanded discussion on my position in an attached PDF, with illustrations of the proposed experiments.